

# EUFF (EUropean Flood Fatalities): A European flood fatalities database since 1980

Olga Petrucci[1], Luigi Aceto[1], Cinzia Bianchi[2], Victoria Bigot[3], Rudolf Brázdil[4,5], Moshe Inbar[6], Abdullah Kahraman[7], Özgenur Kılıç[7], Vassiliki Kotroni[8], Maria Carmen Llasat[9], Montserrat Llasat-Botija[9], Michele
Mercuri[1], Katerina Papagiannaki[8], Susana Pereira[10], Jan Řehoř[4,5], Joan Rossello Geli[11], Paola Salvati[2], Freddy Vinet[3] and José Luis Zêzere[10]

[1]CNR-IRPI National Research Council-Research Institute for Geo-Hydrological Protection, 87036 Rende (Cosenza), Italy
[2]CNR-IRPI National Research Council-Research Institute for Geo-Hydrological Protection, 06128 Perugia, Italy
[3]University Paul Valéry Montpellier 3, 34090 Montpellier, France
[4]Department of Geography, Faculty of Science, Masaryk University, 61137 Brno, Czech Republic
[5]Global Change Research Institute, Czech Academy of Sciences, 60300 Brno, Czech Republic
[6]Department of Geography and Environmental Studies, University of Haifa, 33000 Haifa, Israel
[7]Department of Meteorological Engineering, Faculty of Aeronautics and Astronautics, Samsun University, Ondokuzmayis, Samsun
55420, Turkey
[8]Institute of Environmental Research and Sustainable Development, National Observatory of Athens, 15236 Athens, Greece
[9]Department of Applied Physics, University of Barcelona, 08028 Barcelona, Spain
[10]Centro de Estudos Geográficos, Instituto de Geografia e Ordenamento do Território, Universidade de Lisboa, 1600-276 Lisbon, Portugal
[11]Grup de Climatologia, Hidrologia, Riscs i Paisatge, Universitat Illes Balears, 07122 Palma de Mallorca, Spain

*Correspondence to*: Olga Petrucci (olga.petrucci@irpi.cnr.it)



**Abstract.**

Despite the current developments in flood forecasting and emergency management, floods still consist a significant threat to people and properties. At a national level in Europe, data on flood fatalities are fragmentary and they are mainly focused on death toll, without providing further details regarding victims' characteristics or the circumstances under which the deadly events have taken place. However, such details could enlighten us on what happened wrong when there was a victim due to a flood, and what measures should be taken in order to avoid similar events in the future. This paper presents the *EUFF 2020* database (EUropean Flood

Fatalities-FF) (EUropean Flood Fatalities-FF) (https://doi.org/10.4121/uuid:489d8a13-1075-4d2f-accb-db7790e4542f, Petrucci et al., 2020) which collects data from 2483 flood deadly cases occurred in a 39-year period (1980–2018) in 8 countries and 9 (two belong in Spain) study areas (Czech Republic, Israel, Italy, Turkey, Greece, Portugal, South France, Catalonia and Balearic Islands). *EUFF 2020* (Petrucci et al., 2020) attempts to shed light on fatal flood events in the Euro-Mediterranean region under various geomorphological and climatic settings. The paper presents both regional and super-regional analyses from gender, age, conditions,

activity of fatalities and dynamics of the accidents point of view aiming to contribute to a better understanding of the population exposure to floods phenomena. The historical research, which was carried out by using local documentary sources, highlights that 64.8% of FF have been due to flood events in which less than 10 people were killed. Due to this relatively small number of losses, these events have not been recorded in the international disaster databases. Flood events causing single and multiple fatalities occurred throughout the period of our analysis without showing any particular remarkable trend. Data confirm that victim's gender

only, is not a *de facto* driver of social vulnerability. In addition, females flood fatalities are quantitatively more than males. Males' vulnerability depends on a stronger exposure to floods, due to the higher proportion of males driving vehicles, doing outdoors working activities and sometimes undertaking risky actions. The majority of fatalities are people in their most productive working age (between 30 and 64 years old), who are exposed to floods outdoor while heading from home to work or vice-versa. Elderly people (in status of retirement) seem to be more frequently affected while being indoor, trapped by the flood in their premises, while

adults and children are dragged outdoors. Driving car or any other kind of vehicles are the most frequent conditions of victims in all studied areas, for both males and females, as widely stated in literature. The *EUFF 2020* database can be extended spatially and temporally, and it represents a European database of high scientific and practical potential for further use in various scientific disciplines. We hope, *EUFF 2020* database will further motivate researchers to enrich with even more data on flood fatalities from their home countries. Spatial extension will allow the comparison of local frameworks in broader European scale and the

identification of useful general and local features of risk management and educational campaigns. We believe that the followed pan-European approach, frames the anticipation of flood fatality risk into a broader context, promising benefit for diverse scientific disciplines and contributing to public policies and civil protection campaigns in order to reduce the number of floods' fatalities in the future.



## 1  Introduction

Between 1995–2015, floods represent 47% of the climate-related disasters (Wahlstrom and Guha-Sapir, 2015). The expected increase in floods frequency and magnitude due to climate change (Trenberth, 2011), and the resulted increasing concentration of human activities and people around areas which are prone to floods (IPCC, 2014), make them one of the most important threats for those communities. That was mostly evident during some past catastrophic floods in Europe (Barredo, 2007). Moreover, Blöschl et al. (2020) showed that the period 1990–2016 represent one of the most flood-rich periods in Europe, being exceptional in terms of extent, flood seasonality and air temperatures if compared to similar past flood-rich periods (over the past 500 years).

*Flood fatalities* (FFs), i.e. people who lost their life directly or indirectly during floods, represent the most tragic side of this natural disaster. There are several publications analysing factors that influence the vulnerability of individuals to flooding, mainly related to gender, age, activity and risk taking behaviour, in different geographical and socioeconomic frameworks (Alderman et al., 2012; Fiala, 2017; Lowe et al., 2013; Pereira et al., 2017; Rufat et al., 2015; Špitalar et al., 2014; Brázdil et al., 2019).

These studies are mainly based on specific databases, which in several countries are not even available, and thus they must be created for that specific purpose. At a national level, there are few examples of official FFs databases. In USA for example, the Governmental database *"Storm Data"* is updated by the National Climatic Data Centre (Sharif et al., 2012). The Australian PerilAUS is the database of historical natural hazard impacts, containing FFs that occurred between 1900 and 2015, and critical information such as age, gender, and actions causing death (Coates et al., 2014). At a European level, there are no "official" databases collecting FFs data. The first experiment that was carried out in Mediterranean environment by a multinational research group, is the MEFF (MEditerranean Flood Fatalities) database, which includes FFs occurred on a 36-year period (1980–2015) in five Mediterranean study areas (Petrucci et al., 2019b; Vinet et al., 2019).

Global databases such as NATHAN (Natural Hazards Assessment Network) of the reinsurance company Munich Re, or the EM-DAT (Emergency Events Database) from the Centre for Research on the Epidemiology of Disasters of the Université Catholique de Louvain, are very useful to identify major disasters and extract interesting comparative statistics and useful information. Nevertheless, they contain recordings only for the major catastrophic events and have built-in bias due to the use of indirect sources, or they do consider only the information provided by some insurance companies that do not cover the entirely affected regions (Llasat et al., 2013). Moreover, in these databases the location of FFs is characterized by low spatial resolution.

Flood fatalities databases also allow flood risk scenarios to be developed, taking into account progress and changes in the ongoing lifestyle. Since 1960, the increasing number of cars contributed not only to people's mobility, but also to theirs' increased exposure to flood events, given that people in their cars are more vulnerable (Petrucci et al., 2017; Petrucci and Pasqua, 2012; Sharif et al., 2012) or if they are personnel working for State Emergency Services/Agencies (Ahmed et al., 2020). An interesting tendency which was detected in Australia between 2000–2015 is the dramatic increase of 4WD vehicles drivers' death (Haynes et al., 2016) in their desperate attempt to reach their own or friends' home (Franklin et al., 2017). Some authors suggest that even if most people are aware of the risk involved, the depth and / or the speed of the water might take them by surprise (Coates et al., 2014). It is also noted that even if most drivers may identify the potential risk, they do fail to personalize it, believing that it does not apply to themselves (Gissing et al., 2016; Pearson and Hamilton, 2014), making them impatient and thinking that they are invincible (Franklin et al., 2014) and untouchable. The circumstances can lead to the identification of different types of loss of life (*victims in cars, victims in collapsed buildings, professional rescuers, voluntary rescuers, visitors, observers, victims when structural measures collapsed, victims of non-structural measures*, and *others*) also highlighting the groups of people exhibiting dangerous behaviours and taking unnecessary risks (Špitalar et al., 2020).

The current paper presents the European Flood Fatalities Database (*EUFF 2020*), namely the catalogue of FFs that occurred in nine study areas located in eight countries during a 39-year period (1980–2018). The term "European" is ambitious given that *EUFF 2020* deals with only eight countries; nevertheless, we consider it as the beginning of a larger database that could be supplemented with data from more European countries.



*EUFF 2019* has been presented in a previous paper describing the database and main results in an aggregated form (Petrucci et al., 2019a). The present work is based on an updated version of that database and analyses the number of flood fatalities per event, and the understanding of the relationships between gender and age of victims and the other variables collected, in order to highlight specific vulnerability factors.

Section 2 describes the methodology used to collect data, introduces study areas and presents the structure of the database and its completeness. Section 3 presents the results obtained from data elaboration while Section 4 contains the discussion of the EUFF 2020 potential. Finally, Section 5 provides information regarding the availability of database and Section 6 presents the main conclusions.

## 2 EUFF 2020 databases

### 2.1 Flood fatalities data

The methodological approach is based on a systematic collection of data about floods (*flood events*-FEs) that caused casualties. All cases of fatal floods triggered by rainfall have been included therein, without severity thresholds: *EUFF 2020* (Petrucci et al., 2020) contains all the cases of FEs, independent of the number of FFs per FE.

Such data can be extracted from different types of documentary sources (Brázdil et al., 2012). In our study we took advantage of a common practice (Leal et al., 2018; Papagiannaki et al., 2013; Zêzere et al., 2014), which is the reading of national and local newspapers. In fact, due to their temporal continuity, newspapers allow systematic surveys, which in the frame of the current study were complemented by local sources, different from one study area to another (e.g. reports by rescue services or civil protection agencies). The analysis of newspapers is a long process. It requires the selection of several articles, published either in the daily edition of a single newspaper or in several newspapers of the same day, in order to filter the details and to define the framework in which FFs occurred. Dealing with national newspapers, data gathering must necessarily be performed by researchers understanding the national language, and who are able to easily search throughout several articles and identify the needed information to be included in the database. A typical time sequence of the description of an event may be as the following (from Calabria-Italy):

*October 4, 2018.*

*San Pietro Lametino (Calabria, Italy). Yesterday, the intense rain caused the flood of Cantagalli River, in the province of Lamezia Terme. The river flooded the road of the Ex SIR area and swept away the car driven by Stefania S. (30 yrs). She was going back home after work, at around 19:30, with her children Cristian (7 yrs) and Niccolò (2 yrs). This morning, the car was found near the overpass connecting S. Pietro Lametino to S. Pietro a Maida, with warning lights still activated. Fire brigades, after 14 hours of intense search all night long, found the bodies of Stefania and Cristian.*

*Still missing the little Niccolo': volunteers, police and fire brigades continue tireless to search for him.*

*Source, La Gazzetta del Sud (translated to English)*

### 2.2 Study areas

The *EUFF 2020* database contains information on FF that occurred in 39 years (1980–2018) in nine study areas (BAL: Balearic Islands; CAT: Catalonia; CZE: Czech Republic; SFR: South France; GRE: Greece; ISR: Israel; ITA: Italy; POR: Portugal; TUR: Turkey), located in eight European countries (in case of Spain there are two study areas: Catalonia and Balearic Islands). All countries involved in this project already had either published or not published local databases of the socio-economic impact of different natural hazards. Data sources can be chronicles, books of memory (personal notebooks), weather diaries, newspapers, parliamentary proposals, epigraphic evidence, systematic meteorological/hydrological observations, civil protection reports, professional papers, media websites, radio and TV news, and the community of amateur meteorologists. For BAL, CAT and SFR, the local databases collect only casualties caused by floods. For the remaining study areas, local databases include fatalities caused by different types of natural hazards (Table 1). On the frame of the current study, only flood fatalities have been extracted from

these databases. In order to homogenise the data by filling as much database fields as possible, further research has been performed in each study area by using coeval local newspapers, as *Ultima Hora* for BAL, *La Vanguardia* for CAT, *Rizospastis* for GRE and *Il Corriere della Sera* for ITA. Data assembled in EUFF 2020 are the result of collecting, merging and homogenising regional and

national databases of all the study areas and include further data obtained from recent historical research. In the following, we name 'TOT-Area' the total sum of all study areas (Figure 1).

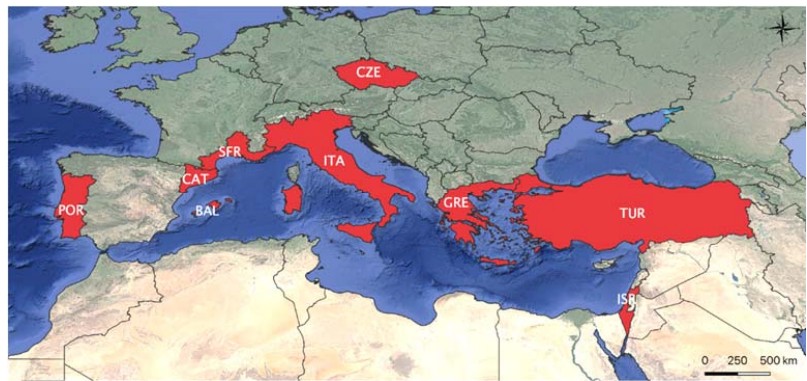

**Figure 1.** Figure 1. Map of the study areas (in red). Legend— BAL: Balearic Islands; CAT: Catalonia; CZE: Czech Republic; SFR: South France; GRE: Greece; ISR: Israel; ITA: Italy; POR: Portugal; TUR: Turkey.

**Table 1.** Main characteristics of the study area and local information sources. ACR: acronyms of the study areas, as reported at the beginning of 2.2 section; Area (km$^2$): surface of the study area; Area (%): surface of the study area as percentage of TOT-Area; Inh (%): inhabitants of the study area as percentage of inhabitants of TOT-Area; PD (Inh/km$^2$): population density; MA (Yrs): mean age of population; Females (%): % of females in the population of the study area. Source for MA and females %: www.Worldometers.info (data 2019), accessed 10 June 2019. *South France includes Languedoc-Roussillon (now part of Occitan region), the Provence-Alpes-Cote d'Azur region and the departments of Drôme and Ardèche

(From Vinet et al., 2019, modified).

| ACR | Area (km$^2$) | Area (%) | Inh (%) | PD (Inh/km$^2$) | MA (Yr) | Females (%) | Existing local database | Type of hazard | References |
|---|---|---|---|---|---|---|---|---|---|
| BAL | 4,492 | 0.3 | 0.6 | 258 | 40 | 50.3 | DB of floods in Balearic Islands (unpublished) | Flood | Grimalt and Rosselló-Geli, 2011 |
| CAT | 32,108 | 2.1 | 3.8 | 235 | 42 | 50.9 | INUNGAMA | Flood | Llasat et al., 2014 |
| CZE | 78,865 | 5.3 | 5.3 | 134 | 43 | 50.9 | Historical-climatological DB of Inst. of Geography, Masaryk (Univ. Brno) | Flood, Flash flood, Windstorm; Convective storm; Lightning; Frost; Snow/glaze-ice; Heat; Other extremes (e.g., avalanche) | Brázdil et al. 2019 |
| SFR | 53,874 | 3.6 | 3.6 | 117.0 | 42 | 51.9 | DB of Department of Geography, University P. Valéry (UMR GRED Lab.) | Flood | Boissier, 2013; Vinet and Boissier, 2016 |
| GRE | 131,957 | 8.8 | 5.4 | 82 | 45 | 50.8 | DB of National Observatory of Athens | Flood, Flash flood, Hail, Snow/frost, Tornado, Windstorm, Heat wave, Lightning | Papagiannaki et al., 2013 |
| ISR | 22,072 | 1.5 | 4.2 | 378 | 31 | 50.3 | DB of Natural hazards in Israel (unpublished) | Earthquake, Flood, Landslide, Drought, Forest fire | --- |
| ITA | 301,338 | 20.1 | 30.5 | 201 | 48 | 51.3 | IRPI (Research Institute for Geo-Hydrological Protection) DB | Flood, Landslide | Petrucci and Pasqua, 2012; Aceto et al., 2017; Petrucci et al., 2017; Salvati et al., 2018 |



| POR | 92,212 | 6.1 | 5.2 | 111 | 46 | 52.7 | DISASTER DB | Flood, Landslide | Zêzere et al., 2014; Pereira et al., 2016, 2017 |
| TUR | 783,562 | 52.2 | 41.4 | 105 | 32 | 50.8 | Turkish Severe Weather DB | Tornado, Hail, Wind, Flood, Lightning | Kahraman et al., 2015; Kahraman and Markowski, 2014; Tilev-Tanriover et al., 2015 |

## 2.3 Database structure

The descriptions of flood events, as identified and selected from data sources (both local databases and additional documentary sources), were used to fill in the *EUFF 2020* fields. Database structure was designed to allow the translation of the flood event

descriptions in a well-defined and restricted number of options, listed in dropdown menus in order to facilitate database compilation. The structure of *EUFF 2020* is detailed in two tables: Table 2 reports the variables used to define the location and time of FE including victim profile, while Table 3 describes the variables used to define the flood-victim interaction.

The database contains the following fields:

➤ PRIMARY KEY is an integer number that allows the univocal identification of each record by means of the FATALITY_ID.

Each record contains data about a single FF, clustered in different sections.

**Table 2.** EUFF **2020** database: variables defining location of accident, time of accident and victim profile.

| | *Variable* | *Type* | *Format* | *Description* |
|---|---|---|---|---|
| PRIMARY KEY | FATALITY_ID | Number | Int | Univocal incremental primary key of the database |
| LOCATION OF ACCIDENT | COUNTRY | Text | String | |
| | REGION | Text | String | |
| | MUNICIPALITY | Text | String | |
| | PREFECTURE | Text | String | |
| | LATITUDE | Decimal degree | Float | Projection: *WGS84 (EPSG: 4326)* |
| | LONGITUDE | Decimal degree | Float | Projection: *WGS84 (EPSG: 4326)* |
| | LOC_ACCURACY | Text | String | Assessment of coordinates accuracy: *LOW/HIGH* |
| TIME OF ACCIDENT | DATE | Date | Date | dd/mm/yyyy |
| | HOUR | Time | String | *Sunrise: 05:00*  *Late afternoon: 16:00* *Early morning: 06:00*  *Evening: 18:00* *Morning: 08:00*  *Late evening: 20:00* *Late morning: 10:00*  *Night: 22:00* *Noon: 12:00*  *Late night: 01:00* *Afternoon: 13:00* |
| | HOUR_ACCURACY | Text | Int | Assessment of hour accuracy: *LOW/HIGH* |
| VICTIM PROFILE | AGE | Number | String | Age of fatality |
| | AGE_STRING | Text | String | *0 − 14 yrs: child* *15−29 yrs: boy/girl* *30 − 49 yrs: young adult* *50 − 64 yrs: adult* *>65 yrs elderly* |
| | GENDER | Text | | *Male: M; Female: F* |
| | RESIDENCY | Text | String | *Resident/Not resident/Tourist* |





➢ LOCATION OF ACCIDENT describes the place where a person was caught up in a flood that eventually caused his/her death.

It contains the following elements: COUNTRY, REGION, MUNICIPALITY and PREFECTURE. If the coordinates in WGS 84 for the exact point where the accident occurred are available, these were also included in the fields LATITUDE and LONGITUDE, and the field LOC_ACCURACY was marked as HIGH. In the cases in which the exact point was not available, LATITUDE and LONGITUDE contain the coordinates of the centroid of MUNICIPALITY, if available, or alternatively of the PREFECTURE or REGION where the accident occurred, and LOC_ACCURACY is marked as LOW.

➢ TIME OF ACCIDENT contains the date in which the accident occurred, in the format dd (day), mm (month) and yy (year). For those cases where the exact hour of the accident was available, that was included in the field HOUR, and the field HOUR_ACCURACY was marked as HIGH. If the hour is reported as a textual description, HOUR_ACCURACY is marked as LOW and the textual description is converted in hours according to the values reported in Table 2.

175                                  **Table 3.** EUFF 2020 database: variables defining flood-victim interaction.

| *Variable* | *Type* | *Format* | *Description* | *Description* |
|---|---|---|---|---|
| VICTIM_CONDITION | Text | String | *By bicycle*<br>*By boat*<br>*By bus*<br>*By car*<br>*By caravan* | *By tractor*<br>*By truck*<br>*By van*<br>*Laying*<br>*Standing* |
| VICTIM_ACTIVITY | Text | String | *Traveling*<br>*Recreational activities*<br>*Rescuing someone*<br>*Sleeping* | *Working*<br>*Hunting*<br>*Fishing* |
| ACCIDENT_PLACE | Text | String | *Public/private building*<br>*Bridge*<br>*Campsite/tent*<br>*Riverbed/riverside*<br>*Tunnel/underpass* | *Countryside*<br>*Ford*<br>*Recreation area*<br>*Road*<br>*Bungalow* |
| ACCIDENT_DYNAMIC | Text | String | *Blocked in a flooded room*<br>*Caught in a bridge collapse*<br>*Caught in a road collapse*<br>*Caught in a building collapse* | *Dragged by water/mud*<br>*Fallen into the river*<br>*Surrounded by water/mud*<br>*Hit* |
| DEATH_CAUSE | Text | String | *Collapse/heart attack*<br>*Drowning*<br>*Hypothermia*<br>*Electrocution* | *Poly-trauma*<br>*Poly-trauma and suffocation*<br>*Suffocation* |
| PROTECTIVE_BEHAVIOUR | Text | String | *Climbing trees*<br>*Driving to avoid danger*<br>*Getting on roof/upper floor*<br>*Getting out of car* | *Getting out of buildings*<br>*Grabbing on to someone/something*<br>*Moving to safer place*<br>*Getting on the car roof* |
| HAZARDOUS_BEHAVIOUR | Text | String | *Check damage during flood*<br>*Driving on roads closed by police*<br>*Fording rivers*<br>*Refuse evacuation*<br>*Trying to rescue animals* | *Refuse warnings*<br>*Staying on bridges*<br>*Staying on river banks*<br>*Trying to save vehicles*<br>*Trying to save belongings* |

➢ VICTIM PROFILE includes the age, gender and residency of FFs. AGE is filled as a number when available. In the majority of cases, the age is a textual description (i.e. *and elderly man*), that is transcribed in the field AGE_STRING. In the cases where age in numerical format was available, the AGE_STRING field was also filled with age class according to age ranges reported



in Table 3. For example, a fatality referred as CHILD was classified in the range 0–14 years. The GENDER, if available, is reported as M (males) or F (females). The field RESIDENCY classifies the victim as RESIDENT or NOT RESIDENT in the place where the accident occurred, or as TOURIST visiting the area.

➤ FLOOD-VICTIM INTERACTION is considered in seven sub-sections: VICTIM_CONDITION, VICTIM_ACTIVITY, ACCIDENT_PLACE, ACCIDENT_DYNAMIC, DEATH_CAUSE, PROTECTIVE_BEHAVIOUR, and

HAZARDOUS_BEHAVIOUR, which are filled by an appropriate item of the lists in the column *Description* of Table 3.

### 2.4    Database completeness

The first version of EUFF presented in 2019 by Petrucci et al. (2019a), contained 2466 FFs that occurred in nine study areas between 1980–2018. The present work is based on an updated version of that database, which, which, according to the year of updating, is named EUFF 2020. Improvements from EUFF 2019 to EUFF 2020 are summarised in Table 4. As can be noticed, the difference

between the number of FFs in the two versions of the database is relatively low (17 FFs). This is expected since the number of fatalities in newspapers is the most frequently information reported therein. Also, considering that this research has been carried out by a systematic surveying of newspapers, it is unlikely that large number of fatalities can be remained unnoticed and could emerge from further research. The small increasing of FFs from EUFF 2019 to EUFF 2020 essentially stems from supplementary research on short periods for which data were missing. For example, it is not rare that printed collections of old newspapers owned by libraries

and newspaper libraries can be affected by gaps, due i.e. to deterioration of some edition caused by humidity in the library's premises. In these cases, if digital versions or private collections of these missing editions become available, the gaps can be filled, and, if in that period some FFs occurred, the total number of FFs can be updated.

On the other hand, in EUFF 2020 there is a larger increase in the availability of variables such as ACCIDENT-PLACE, RESIDENCY, VICTIM_CONDITION and ACCIDENT_DINAMYC. This depends mainly on the current availability of coeval

data sources that, for already counted FEs, allows the identification of more details not initially available in the documentary sources analysed during EUFF 2019. In these cases, the absolute number of FFs did not increase but the completeness of the database has been critically improved, thus contributing a more realistic framework and a more robust basis for statistical data elaboration and analysis.

**Table 4.** Data included in EUFF (2019) and EUFF (2020), and increases as numbers (#) and percentage (%) of the FFs in EUFF (2020). The diagram on the right represents the numerical increasing of data available for each variable.

| | EUFF (2019) | EUFF (2020) | Increase # | Increase % |
|---|---|---|---|---|
| **FLOOD FATALITIES** | 2466 | 2483 | 17 | 0.68 |
| **FLOOD EVENTS** | 812 | 847 | 35 | 1.41 |
| **AGE** | 1588 | 1602 | 14 | 0.56 |
| **GENDER** | 1936 | 1953 | 17 | 0.68 |
| **RESIDENCY** | 1482 | 1666 | 184 | 7.41 |
| **VICTIM_CONDITION** | 795 | 978 | 183 | 7.37 |
| **VICTIM_ACTIVITY** | 789 | 901 | 112 | 4.51 |
| **ACCIDENT_PLACE** | 1270 | 1531 | 261 | 10.51 |
| **ACCIDENT_DYNAMIC** | 1838 | 2020 | 182 | 7.33 |
| **DEATH_CAUSE** | 1994 | 2001 | 7 | 0.28 |
| **PROTECTIVE_BEHAVIOUR** | 130 | 163 | 33 | 1.33 |
| **HAZARDOUS_BEHAVIOUR** | 262 | 325 | 63 | 2.54 |

Currently, EUFF 2020 contains 2483 flood fatalities (Figure 2). Half of them occurred in Turkey (50.1%) followed by Italy (16.4%) and south France (11.0%). The remaining 22.6% concern the remaining six areas.

Table 5 summarises the data collected in TOT-Area and in the individual study areas. For each variable, we reported the data available expressed as proportion of FFs in each study area.

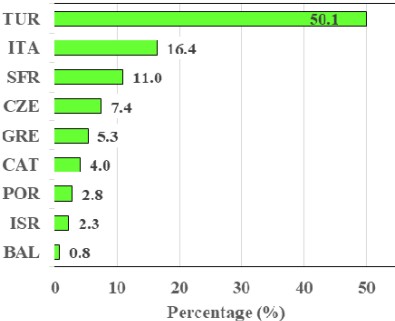

**Figure 2.** Percentage of flood fatalities occurred in the period 1980–2018 in each study area.

**LOCATION OF ACCIDENT**, REGION is available for 99.6% of FFs; MUNICIPALITY for 47.4% and PREFECTURE for 99.5%
of cases, while high accuracy LATITUDE/LONGITUDE coordinates are available in 27.4% of FFs. The most complete data on the location of the accident concerns CAT and POR, where these variables are available for all FFs.

**TIME OF ACCIDENT**, DATE is available for 100% of FFs in all the study areas.

Concerning **VICTIM PROFILE**, we are aware of the AGE of 64.5% of FFs: the highest proportions pertain to BAL (100%) and ITA (99.3%). GENDER is available for 78.6% of FFs: most complete data pertains to GRE and ITA. RESIDENCY is available for
67.1% of FFs, while the highest percentages concern GRE and CAT. Concerning **FLOOD-VICTIM INTERACTION**, we are aware of VICTIM_CONDITION for 39.3% of FFs: the highest completeness appears for CAT. VICTIM_ACTIVITY is available for 37.7% of FFs, while completeness is highest for CAT. ACCIDENT_PLACE is available for 61.7% of FFs and the most complete data on this variable pertains to ITA. ACCIDENT_DYNAMIC is available for 81.4% of FFs, and the highest percentage of data pertains to GRE. DEATH_CAUSE is known for 80.6% of FFs. In a few number of cases, PROTECTIVE_BEHAVIOUR (6.6%)
and HAZARDOUS_BEHAVIOUR (13.1%) were also detected.

**Table 5.** Proportions of data (%) collected for each variable in EUFF 2020 with respect to flood fatalities occurred in TOT-Area (in red) and in each study area (in black).

| | LOCATION OF ACCIDENT | | | | TIME OF ACCIDENT | VICTIM PROFILE | | | FLOOD-VICTIM INTERACTION | | | | | | |
|---|---|---|---|---|---|---|---|---|---|---|---|---|---|---|---|
| | FFs | REGION | MUNICIPALITY | PREFECTURE | LAT/LON | HOUR | AGE | GENDER | RESIDENCY | VICTIM_CONDITION | VICTIM_ACTIVITY | ACCIDENT_PLACE | ACCIDENT_DYNAMIC | DEATH_CAUSE | PROTECTIVE_BEHAVIOUR | HAZARDOUS_BEHAVIOUR |
| TOT-Area | 2483 | 99.6 | 47.4 | 99.5 | 27.4 | 28.1 | 64.5 | 78.7 | 67.1 | 39.3 | 37.7 | 61.7 | 81.4 | 80.6 | 6.6 | 13.1 |
| BAL | 20 | 100 | 100 | 100 | 80.0 | - | 100 | 70.0 | 70.0 | 70.0 | 60.0 | 70.0 | 70.0 | 100 | 15.0 | - |
| CAT | 100 | 100 | 100 | 100 | 100 | 53.0 | 90.0 | 92.0 | 87.0 | 87.0 | 77.0 | 94.0 | 97.0 | 99.0 | 14.0 | 50.0 |
| CZE | 183 | 100 | 96.2 | 99.5 | - | 6.6 | 74.9 | 92.9 | 16.9 | 16.9 | 27.9 | 66.1 | 49.2 | 89.6 | 1.6 | 22.4 |
| SFR | 273 | 100 | 100 | 100 | - | 50.2 | 97.1 | 96.7 | 81.7 | 81.7 | 56.4 | 94.1 | 76.2 | 99.3 | 12.1 | 22.0 |
| GRE | 132 | 100 | 100 | 100 | 72.7 | 96.2 | 89.4 | 98.5 | 79.5 | 79.5 | 75.0 | 94.7 | 97.0 | 100 | 8.3 | 23.5 |
| ISR | 56 | 96.4 | - | 94.6 | - | 23.2 | 26.8 | 28.6 | 42.9 | 42.9 | 35.7 | 32.1 | 58.9 | 80.4 | - | - |
| ITA | 407 | 100 | 100 | 100 | 98.0 | 53.3 | 99.3 | 97.5 | 78.6 | 78.6 | 67.1 | 95.8 | 96.1 | 99.3 | 83.9 | 20.4 |
| POR | 69 | 100 | 100 | 100 | 100 | 33.3 | 88.4 | 94.2 | 73.9 | 73.9 | 44.9 | 88.4 | 84.1 | 11.6 | 2.0 | 47.8 |
| TUR | 1243 | 99.3 | - | 99.3 | 0.1 | 7.7 | 39.6 | 64.8 | 9.7 | 9.7 | 17.5 | 36.3 | 80.5 | 69.0 | 63.8 | 2.2 |



## 3    Results

### 3.1    Flood events and flood fatalities

The number of FFs caused by a single FE can be a proxy of the severity of the flood. Generally, the larger the number of FFs, the higher the severity of the FE. Basic assumption is that a FE is a flood that caused the death of at least one or more people in a given DATE and REGION. Once the region or the date change, the FFs are assigned to another FE. Using this criterion, in 1980–2018 period we counted 847 FEs causing 2483 FFs in TOT-Area (Table 6), i.e. 2.9 casualties per event, on average.

**FEs** reached the highest number in TUR (328), ITA (168) and CZE (92).

**FFs** shows the highest numbers in TUR (1243), ITA (407) and SFR (273).

**FEs per year** has the highest value in TUR (26 FE in 1981), followed by ITA (13 FE in 2011) and CZE (12 FE in 1997). After TUR, with a mean number of 8.9 FEs per year, high values pertain to both ITA (5.1) and SFR (3.3). The modal value of FEs per year is 10 in TUR, and it ranges between one and two in the other areas.

**FFs per year** reaches the maximum value in TUR (157 FFs in 1995), followed by SFR (56 FFs in 1992). The highest mean number

of FFs per year pertains to TUR (33.6), followed by ITA (12.3).

**FFs per FE** show the maximum value for TUR, where one FE in 1995 caused 74 FFs; the second value pertains to ITA (46 FFs during a FE occurred in 1994). The mean number of FFs caused by a single FE changes among study areas: TUR presents the highest value (3.8), while the lowest pertains to BAL (1.4 FF per FE). Based on EUFF 2020 data, we set eight classes of number of Flood Fatalities per Flood Event (FFs per FE).

**Table 6.** Number of Flood Events (FEs) and Flood Fatalities (FFs). The column Year reports the year in which the maximum value has been achieved (V.Y.: Various years).

|  | FEs | FFs | FEs per year | | | | FFs per year | | | | FFs per FE | | | |
|---|---|---|---|---|---|---|---|---|---|---|---|---|---|---|
|  |  |  | Max | Year | Mean | Modal value | Max | Year | Mean | Modal value | Max | Year | Mean | Modal value |
| BAL | 6 | 20 | 1 | V.Y. | 1.0 | 1 | 13 | 2018 | 3.3 | 1 | 13 | 2018 | 1.4 | 1 |
| CAT | 46 | 100 | 6 | 1987 | 1.7 | 1 | 16 | 1982 | 3.7 | 1 | 12 | 1982 | 2.7 | 1 |
| CZE | 92 | 183 | 12 | 1997 | 3.2 | 1 | 48 | 1997 | 6.3 | 1 | 32 | 1997 | 2.0 | 1 |
| SFR | 85 | 273 | 9 | 2002 | 3.3 | 1 | 56 | 1992 | 10.5 | 2 | 42 | 1992 | 3.2 | 1 |
| GRE | 64 | 132 | 6 | V.Y. | 2.8 | 1 | 25 | 2017 | 5.7 | 1 | 24 | 2017 | 2.1 | 1 |
| ISR | 21 | 56 | 4 | V.Y. | 1.9 | 1 | 12 | 2018 | 5.1 | 2 | 12 | 2018 | 2.7 | 1 |
| ITA | 168 | 407 | 13 | 2011 | 5.1 | 2 | 50 | 1994 | 12.3 | 9 | 46 | 1994 | 2.9 | 1 |
| POR | 37 | 69 | 6 | 1996 | 2.1 | 1 | 19 | 1983 | 3.8 | 1 | 18 | 1983 | 1.9 | 1 |
| TUR | 328 | 1243 | 26 | 1981 | 8.9 | 10 | 157 | 1995 | 33.6 | 3 | 74 | 1995 | 3.8 | 1 |
| TOT-Area | 847 | 2483 | 41 | 2014 | 21.7 | 21 | 174 | 1995 | 63.7 | 70 | 74 | 1995 | 2.9 | 1 |

In Table 7 the percentage of FFs across the different classes of FEs is shown, while Figure 3 represents the frequency of FFs in these classes. Looking to Table 7, more than 60% of FFs happened during FEs responsible for the loss of less than 10 people.

Particularly, 27.1% of FFs happened during FEs killing 2–4 people, 19.3% due to FEs that killed a single person, and 18.3% due to FEs killing between 5−10 people. FEs killing larger number of people are numerous mainly in TUR. Highly deadly events (more than 40 casualties) have been recorded only in TUR, SFR and ITA.





**Table 7.** Number of flood fatalities (FFs) per flood event (FE) as percentage of total FFs.

| FFs per FE | BAL | CAT | CZE | SFR | GRE | ISR | ITA | POR | TUR | TOT |
|---|---|---|---|---|---|---|---|---|---|---|
| 1 | 0.2 | 1.2 | 2.4 | 2.1 | 1.8 | 0.3 | 4.1 | 1.2 | 6.0 | 19.3 |
| 2–4 | 0.1 | 1.2 | 3.0 | 2.0 | 1.7 | 1.1 | 5.0 | 0.4 | 12.6 | 27.1 |
| 5–10 | - | 1.1 | 0.2 | 1.9 | 0.9 | 0.3 | 3.2 | 0.4 | 10.2 | 18.3 |
| 11–20 | 0.5 | 0.5 | 0.4 | 2.5 | - | 0.5 | 2.3 | 0.7 | 5.0 | 12.4 |
| 21–30 | - | - | - | 0.8 | 1.0 | - | - | - | 2.2 | 4.0 |
| 31–40 | - | - | 1.3 | - | - | - | - | - | 2.7 | 3.9 |
| 41–50 | - | - | - | 1.7 | - | - | 1.8 | - | 3.7 | 7.2 |
| >50 | - | - | - | - | - | - | - | - | 7.7 | 7.7 |
| TOT | 0.8 | 4.0 | 7.4 | 11.0 | 5.3 | 2.3 | 16.4 | 2.8 | 50.1 | 100 |

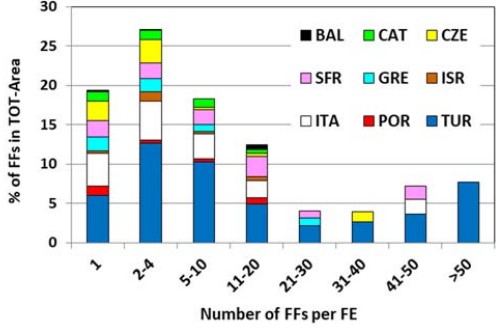

**Figure 3**. Percentage of flood fatalities (FFs) in TOT-Area, clustered according to the number of FFs per FE.

Figure 4 illustrates the annual distribution of FFs and the maximum annual number of FFs per FE, not only in TOT-Area but also

in the individual regions. The highest number of FFs per year (174) was recorded in 1995, due to different FEs affecting TUR (157 FFs), ITA (12 FFs), SFR (2 FFs), CZE (2 FFs) and CAT (1 FF). TUR shows the highest frequency of FEs per year: in 37 out of totally 39 years, any FF has been occurred. High frequency of FEs also appears for ITA (FEs occurrence in 33 out of 39 years) and CZE (FEs in 29 out of 39 years). For TOT-Area, the general trend seems to be quite stable, in terms of both FFs per year and maximum number of FFs per FE. The **number of FFs per year** (red dotted lines) shows an upward trend in BAL, CZE, SFR, GRE

and ITA, it tends to decrease in CAT, POR and TUR, while it remains stable over time for ISR. The **maximum number of FFs per FE** (black dotted lines) shows an upward annual trend in BAL, GRE and SFR, it slightly decreases in CAT, POR and TUR, while it remains stable in CZE, ITA, and ISR cases.



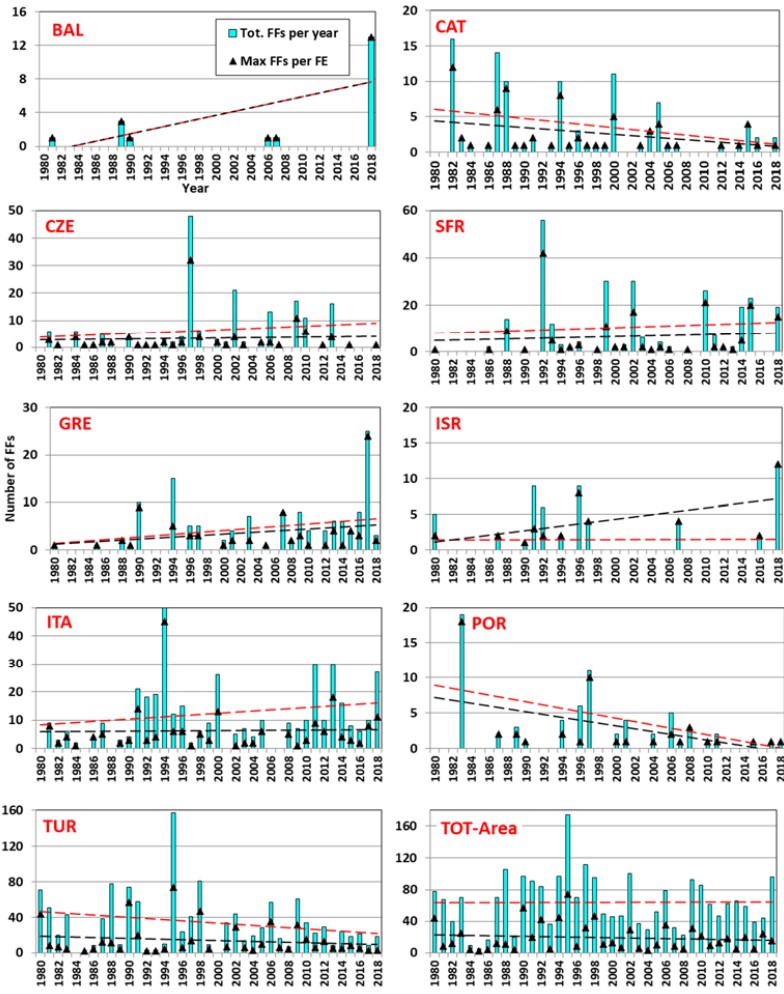

**Figure 4.** Number of flood fatalities (FFs) per year (bars) and maximum number of FFs per FE (triangles) in the individual areas and TOT-Area in the 1980–2018 period. Red dotted line is the linear trend of FFs per year, black dotted line is the trend in the maximum number of FFs per FE.

## 3.2 Gender of flood fatalities

**Gender** is known for 1953 FFs (i.e., 78.6% of total FFs): 47.0% of FFs have been males, 31.7% females, and for the remaining 21.3% information on gender was missing (Table 8). Among the individual areas, completeness of this information ranges between 98.5% (GRE) and 28.6% (ISR) of FFs that occurred in the given area (Table 5). In all areas studied, males FFs are more in absolute numbers than females.





**Table 8.** Gender of flood fatalities according to their age, residency, condition, and activity.

| GENDER by study area | | | |
|---|---|---|---|
| | **Females** | **Males** | **Gender unknown** | **Total** |
| BAL | 0.2 | 0.3 | 0.2 | **0.8** |
| CAT | 1.4 | 2.3 | 0.3 | **4.0** |
| CZE | 1.8 | 5.1 | 0.5 | **7.4** |
| SFR | 4.6 | 6.1 | 0.4 | **11.0** |
| GRE | 1.7 | 3.5 | 0.1 | **5.3** |
| ISR | 0.1 | 0.5 | 1.6 | **2.3** |
| ITA | 6.1 | 9.9 | 0.4 | **16.4** |
| POR | 1.0 | 1.7 | 0.2 | **2.8** |
| TUR | 14.9 | 17.5 | 17.6 | **50.1** |
| **Total** | **31.7** | **47.0** | **21.3** | **100** |

| GENDER and AGE | | | |
|---|---|---|---|
| | **Females** | **Males** | **Gender unknown** | **Total** |
| Child | 4.4 | 5.0 | 2.9 | **12.3** |
| Boy/girl | 3.8 | 5.3 | 0.3 | **9.5** |
| Young adult | 5.2 | 9.1 | 0.8 | **15.0** |
| Adult | 5.3 | 11.1 | 0.3 | **16.6** |
| Elderly | 5.7 | 5.3 | 0.1 | **11.1** |
| Age unknown | 7.3 | 11.2 | 16.9 | **35.5** |
| **Total** | **31.7** | **47.0** | **21.3** | **100** |

| GENDER and RESIDENCY | | | |
|---|---|---|---|
| | **Females** | **Males** | **Gender unknown** | **Total** |
| Resident | 20.7 | 27.3 | 10.6 | **58.5** |
| Not resident | 0.6 | 2.3 | 0.2 | **3.1** |
| Tourist | 2.3 | 3.0 | 0.2 | **5.4** |
| Residency unknown | 8.1 | 14.4 | 10.4 | **32.9** |
| **Total** | **31.7** | **47.0** | **21.3** | **100** |

| GENDER and VICTIM CONDITIONS | | | |
|---|---|---|---|
| | **Females** | **Males** | **Gender unknown** | **Total** |
| Car | 5.5 | 11.9 | 1.1 | **18.5** |
| Standing | 4.7 | 7.8 | 0.9 | **13.4** |
| Other vehicles | 1.2 | 2.3 | 0.8 | **4.3** |
| Laying | 1.2 | 0.7 | 0.1 | **2.1** |
| Boat | 0.2 | 0.8 | 0.1 | **1.1** |
| Condition unknown | 18.8 | 23.4 | 18.3 | **60.6** |
| **Total** | **31.7** | **47.0** | **21.3** | **100** |

| GENDER and VICTIM ACTIVITY | | | |
|---|---|---|---|
| | **Females** | **Males** | **Gender unknown** | **Total** |
| Traveling | 6.5 | 12.8 | 1.9 | **21.2** |
| Working | 0.9 | 4.6 | 0.5 | **6.1** |
| Recreational activities | 1.2 | 2.5 | 0.7 | **4.4** |
| Sleeping | 1.4 | 1.0 | 0.2 | **2.5** |
| Rescuing someone | 0.4 | 1.6 | - | **2.0** |
| Activity unknown | 21.3 | 24.4 | 18.0 | **63.7** |
| **Total** | **31.7** | **47.0** | **21.3** | **100** |

**Gender and age** information were available for 1492 FFs (60.1%). Adults (16.6%) and young adults (15.0%) have been the two prominent cases. In

Figure 5, the small histogram on top of each figure represents gender and age of FFs, as a percentage from gender and age (reported
in brackets on top of each diagram). The histograms at the bottom of each figure represent gender and age of the population of each
study area, as a percentage of the total population (http://www.populationpyramid.net/, data 2017). For study areas where either the
number of FFs is low (BAL) or data about gender and age are scarce (ISR and TUR), it is not possible to draw any fruitful conclusion.
Among the other study areas, the majority of FFs was either adults or young-adults, i.e., people between 30 and 64 years. In each
age class, females are generally less in absolute numbers than males. Nevertheless, in CAT, CZE, SFR, ITA, and POR, in the class
of elderly females become slightly more than males. This may reflect the age structure of the population in these study areas, where,
among elderly, the prevalence of females on males is stronger than in the rest of age classes (
Figure 5).

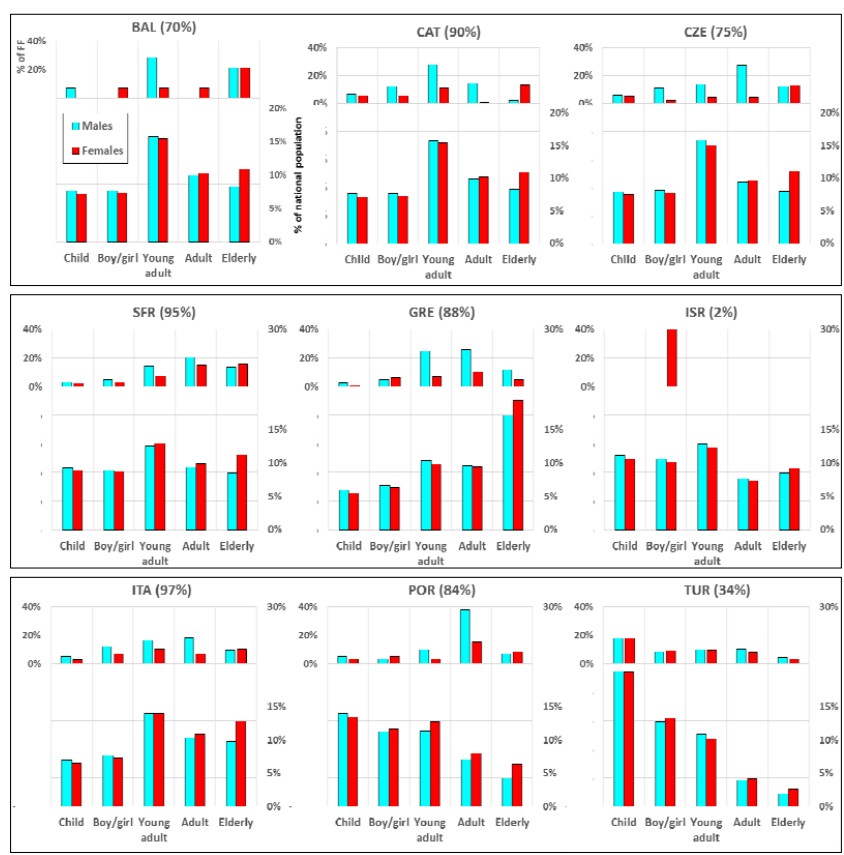

**Figure 5.** Top of figures: gender and age of flood fatalities (FFs) in each study area, expressed as percentage of total flood fatalities for which
data are available (numbers in brackets). Bottom of figures: histograms of gender and age of population in each study area, expressed as
percentage of total population (http://www.populationpyramid.net/, data 2017).

Concerning **gender and residence** of FFs, we have information for 1395 FFs (56.2%) (Table 8). The majority of FFs were residents
in the area where the accident took place, both males and females, i.e. we can make a safe assumption that they were aware of the
local places and roads with high risk for flooding. The percentages of FFs that were either not residents in the place of the accident
(i.e.: being there for work) or tourists are small (3.1% and 5.4%, respectively).

**Gender and victim condition** are known for 903 FFs (36.4%) (Table 8). Data are scarcely available for TUR, CZE and ISR, while
for the remaining study areas, this couple of variables is available for around 70% of FFs (Table 5). Due to the low number of cases,

we grouped *bicycle, bus, caravan, tractor, truck and van*, in the new class "*other vehicles*". *Car* is the most frequent mean of transport in which a deadly even has happened, for both males and females, and in each study area.

**Gender and victim activity** are known for 818 FFs (32.9%), and more complete information concerns CAT, GRE and ITA (Table 8). This information confirms that the majority of FFs, particularly males, were *traveling* (by car or other vehicles), as often mentioned in literature (e.g. Jonkman and Kelman 2005). The second most frequent activity for male FFs was *working*, particularly

in CAT, GRE, POR and CZE. Concerning female FFs, after traveling, the second most frequent activity was *sleeping*, so they were probably involved in the flood in a state of unconsciousness. Male fatalities during hunting and fishing activities, due to their low number have been clustered as 'recreational activities'.

**Gender and accident place** are known for 1371 FFs (55.2%). These data were grouped in four types: A) *Riverbed/riverside, Ford, Bridge*; B) *Road, Tunnel/underpass*; C) *Campsite/tent, Countryside, Bungalow, Recreation area*; D) *Public/private building* (Table

9). In CAT, SFR, GRE and ITA, females in *Public/private building* were more than males (Figure 6). This is in accordance with the societal role of females in south and central European societies, spending more time at home than males, due to their greater charge of work and responsibilities in the care of house and children.

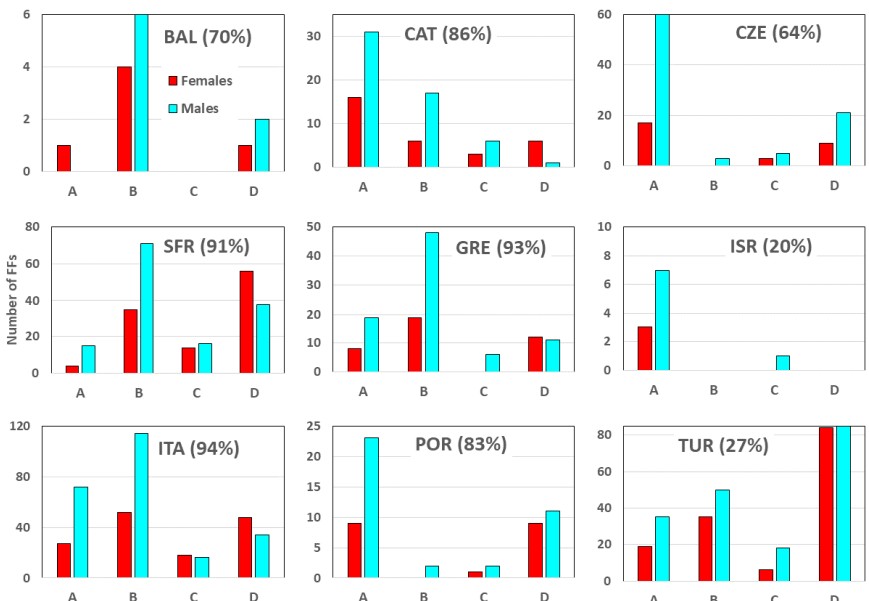

**Figure 6.** Number of flood fatalities (FFs) per gender and accident place in the each study area: A) Riverbed/riverside, Ford, Bridge; B) Road,
Tunnel/underpass; C) Campsite/tent, Countryside, Bungalow, Recreation area; D) Public/private building. Number in brackets are the percentages of FFs for which both gender and accident place are known.

Gender and accident dynamic are known for 1676 FFs (67.5%), and they are largely available for almost all studied areas, except for ISR (Table 9). Accident dynamics was clustered in five groups: A) *Blocked in a flooded room*; B) *Caught in a bridge collapse, Caught in a road collapse,*
*Fallen into the river*; C) *Dragged by water/mud*; D) *Caught in building collapse, Hit*; E) *Surrounded by water/mud*. The most common dynamic was C: *Dragged by water/mud,* for both males and females. Second was group B: *Caught in a bridge collapse, Caught in a road collapse, and Fallen into the river*) for males, and group A: *Blocked in a flooded room* for females. *Blocked in a flooded room* was more numerous for females in ITA, POR, GRE and SFR, while only females died with this dynamic in CAT and CZE. This may depend on gender roles, according which a larger number of females were at home during floods and were unable to effectively face water e.g find their exit from a flooded room and save
themselves.



Table 9. Gender of flood fatalities according to the accident place, accident dynamic, protective and risky behaviours.

| GENDER and ACCIDENT PLACE | | | |
|---|---|---|---|
| | Females | Males | Gender unknown | Total |
| A) Riverbed/riverside, Ford, Bridge | 4.2 | 10.6 | 0.4 | 15.2 |
| B) Road, Tunnel/underpass | 6.1 | 12.5 | 1.9 | 20.5 |
| C) Campsite/tent, Countryside, Bungalow, Recreation area | 1.8 | 2.8 | 0.3 | 5.0 |
| D) Public/private building | 9.1 | 8.2 | 3.8 | 21.0 |
| Accident place unknown | 10.6 | 12.9 | 14.9 | 38.3 |
| **Total** | **31.7** | **47.0** | **21.3** | **100** |
| GENDER and ACCIDENT DYNAMIC | | | |
| | Females | Males | Gender unknown | Total |
| A) Blocked in a flooded room | 5.0 | 4.0 | 0.8 | 9.9 |
| B) Caught in a bridge collapse, Caught in a road collapse, Fallen into the river | 2.3 | 5.0 | 0.3 | 7.6 |
| C) Dragged by water/mud | 16.9 | 27.1 | 9.7 | 53.6 |
| D) Caught in building collapse, Hit | 2.8 | 3.2 | 2.8 | 8.8 |
| E) Surrounded by water/mud | 0.5 | 0.7 | 0.2 | 1.4 |
| Accident dynamic unknown | 4.1 | 7.0 | 7.5 | 18.6 |
| **Total** | **31.7** | **47.0** | **21.3** | **100** |
| GENDER and DEATH CAUSE | | | |
| | Females | Males | Gender unknown | Total |
| A) Collapse/heart attack | 2.8 | 4.1 | 2.6 | 9.5 |
| B) Drowning | 23.0 | 35.9 | 9.7 | 68.5 |
| C) Hypothermia; Electrocution | 0.2 | - | - | 0.2 |
| D) Poly-trauma, suffocation, poly-trauma and suffocation | 0.8 | 1.3 | 0.2 | 2.3 |
| Death cause unknown | 4.9 | 5.6 | 8.9 | 19.4 |
| **Total** | **31.7** | **47.0** | **21.3** | **100** |
| GENDER and PROTECTIVE BEHAVIOUR | | | |
| | Females | Males | Gender unknown | Total |
| A) Climbing trees, Grabbing on to someone/something | 0.1 | 0.6 | - | 0.7 |
| B) Driving to avoid danger, Getting on the car roof, Getting out of car | 1.2 | 2.0 | - | 3.2 |
| C) Getting out of buildings, Moving to safer place, Getting on roof/upper floor | 1.2 | 1.1 | 0.3 | 2.7 |
| Protective behaviour unknown | 29.1 | 43.3 | 21.0 | 93.4 |
| **Total** | 31.7 | 47.0 | 21.3 | 100 |
| GENDER and HAZARDOUS BEHAVIOUR | | | |
| | Females | Males | Gender unknown | Total |
| A) Check damage during flood | 0.1 | 0.5 | - | 0.6 |
| B) Driving on roads closed by police; Fording rivers | 1.5 | 3.4 | 0.1 | 5.0 |
| C) Refuse evacuation; Refuse warnings | 0.2 | 1.6 | 0.1 | 1.9 |
| D) Staying on bridges; Staying on river banks | 0.8 | 1.9 | - | 2.7 |
| E) Trying to save vehicles; Trying to save belongings; Trying to rescue animals | 0.8 | 2.0 | - | 2.9 |
| Hazardous behaviour unknown | 28.3 | 37.6 | 21.1 | 86.9 |
| **Total** | **31.7** | **47.0** | **21.3** | **100.0** |

**Gender and death cause** information is available for 1692 FFs (i.e., 68.1%). We matched these data in four main groups: A) *Collapse/heart attack;* B) *Drowning;* C) *Hypothermia; Electrocution;* and D) *Poly-trauma; Suffocation: Poly-trauma and suffocation.* Except for POR and TUR, data on both gender and cause of death are available for more than 80% of FF (Table 9).





Most commonly, people died due to drowning, and secondly due to collapse/heart attack. The percentages between males and females essentially reflect the proportions males/females among FFS in each study area, without showing any particular trend.

**Gender and protective behaviours** have been detected only for 155 FFs (6.2%). We grouped protective behaviours into three main types (Table 9): A) *Climbing trees, Grabbing on to someone/something*; B) *Driving to avoid danger, Getting on the car roof, Getting out of car*; and C) *Getting out of buildings, Moving to safer place, Getting on roof/upper floor*. Males most frequently attempted a behaviour of group B, while females of group C. Despite the scarcity of data, females behaving to protect themselves counted 8.1% of total females FF, while males were 7.8% of males FF. Nevertheless, if we refer to percentages of protective behaviours to the

total number of cases (155 FFs), the majority have been males (58.7%), mainly showing attitudes of type B.

**Gender and hazardous behaviours** are available for 318 FFs (14.7%). Data are not provided for BAL and ISR (Table 9). However, this does not mean that people behaved responsibly; merely, this information was either not available or not collected. We grouped five types of hazardous behaviour: A) *Check damage during flood*; B) *Driving on roads closed by police; Fording rivers*; C) *Refuse evacuation; Refuse warnings*; D) *Staying on bridges; Staying on river banks*; and E) *Trying to save vehicles; Trying to save*

*belongings; Trying to rescue animals*. The hazardous females represented 10.8% of total females FFs, while for hazardous males this proportion was 20.0% of males FFs. If we refer to the percentages of the total number of hazardous behaviours (318 FFs), males account for 73.2%, while females only 26.7%. From a local point of view, *Driving on roads closed by police* and *Fording rivers* were frequent in CAT, GRE, ITA and SFR. *Check damage during flood* was the most frequent in CZE. *Staying on bridges* and *Staying on river banks* were recorded mainly in POR. *Trying to save vehicles; Trying to save belongings; Trying to rescue animals*

were mainly deadly reasons in SFR and TUR.

### 3.3 Age of flood fatalities

**Age** is known for 1602 FFs (64.5%). This information is fragmentary for ISR and TUR and it is largely available for the other areas (Table 10). Data on both **age and residency** are available for 1193 FFs (48.0%). Residents have been in all age classes. Tourists created a low percentage of FFs, mainly among adults and young adults. Concerning **age and victim condition**, we have information

on 26.9% of FFs. By *car* is common in each class of age, with highest values in young adult and adult classes. The condition *laying* seems more frequent in elderly people. **Age and victim activity** is known for 638 FFs (25.7%) (Table 11). Largest percentage pertains to young adults *traveling*. *Sleeping* is quite common in elderly. A small percentage of FFs, in all the age classes, were doing *recreational activities,* and a small percentage of adults, young adults and boys/girls were *rescuing someone*.

**Age and accident place** information are available for 1004 FFs (40.4%) (Table 11). The largest percentage of FFs (17.7%) were in

*public and private buildings*, and mainly were elderly people. *Road and tunnel/underpass* were the second most frequent case, mainly for adults and young adults. In ITA and GRE, the majority of FFs (essentially younger people) occurred outdoor (*road and tunnel/underpass*), while elderly people are affected indoor (*public/private buildings*). CAT, CZE and POR show a predominance of occurrences outdoor (*riverbed/riverside, ford* and *bridges*) in almost all classes of age, except for the elderly, which have been once again, more frequently affected indoor.

**Age and accident dynamic** data are available for 1250 FFs (50.3%), and *dragged by water/mud* shows the highest percentage in all the age classes, mainly in adult and young adult range of age (Table 11). Among elderly, the second most frequent reason was *blocked in a flooded room*.

**Age and death cause** are available for 1470 FFs (59.2%) (Table 11). *Drowning* killed most people in all the age classes. The second most frequent cause of death was *collapse/heart attack,* again in all age classes. Nevertheless, the relative incidence of this death

cause is slightly higher between children and elderly fatalities.

**Age and protective behaviour** are available for only 116 FFs (4.7%) (Table 11). These behaviours seem relatively more frequent between young adults and adults, especially in the type B (*Driving to avoid danger, Getting on the car roof, Getting out of car*). Boy/girl and elderly classes of age show the lowest frequency of protective behaviour.





**Age and hazardous behaviour** were available for 238 FFs (9.6%), concerning mainly adults and young adults (Table 11). Young adults exhibited more hazardous behaviours in CAT, GRE and ITA (type B: *Driving on roads closed by police; Fording rivers*), and adults in POR and CZE (type B: *Driving on roads closed by police; Fording rivers,* and type C: *Refuse evacuation; Refuse warnings*).

**Table 10.** Age of flood fatalities crosschecked with their age, residency, condition, and activity.

| AGE by study area | | | | | | | |
|---|---|---|---|---|---|---|---|
| | Child | Boy/girl | Young adult | Adult | Elderly | Age unknown | Total |
| BAL | - | 0.1 | 0.2 | 0.2 | 0.3 | - | **0.8** |
| CAT | 0.4 | 0.6 | 1.4 | 0.6 | 0.6 | 0.4 | **4.0** |
| CZE | 0.6 | 0.7 | 1.0 | 1.8 | 1.4 | 1.9 | **7.4** |
| SFR | 0.7 | 0.9 | 2.3 | 3.7 | 3.1 | 0.3 | **11.0** |
| GRE | 0.2 | 0.5 | 1.5 | 1.7 | 0.8 | 0.6 | **5.3** |
| ISR | 0.1 | - | 0.5 | - | - | 1.7 | **2.3** |
| ITA | 1.3 | 3.1 | 4.4 | 4.1 | 3.3 | 0.1 | **16.4** |
| POR | 0.2 | 0.2 | 0.3 | 1.4 | 0.4 | 0.3 | **2.8** |
| TUR | 8.7 | 3.2 | 3.4 | 3.2 | 1.3 | 30.2 | **50.1** |
| **Total** | **12.3** | **9.5** | **15.0** | **16.6** | **11.1** | **35.5** | **100** |
| AGE and GENDER | | | | | | | |
| | Child | Boy/girl | Young adult | Adult | Elderly | Age unknown | Total |
| Females | 4.4 | 3.8 | 5.2 | 5.3 | 5.7 | 7.3 | **31.7** |
| Males | 5.0 | 5.3 | 9.1 | 11.1 | 5.3 | 11.2 | **47.0** |
| Gender unknown | 2.9 | 0.3 | 0.8 | 0.3 | 0.1 | 16.9 | **21.3** |
| **Total** | **12.3** | **9.5** | **15.0** | **16.6** | **11.1** | **35.5** | **100** |
| AGE and RESIDENCY | | | | | | | |
| | Child | Boy/girl | Young adult | Adult | Elderly | Age unknown | Total |
| Resident | 4.7 | 9.7 | 7.9 | 10.4 | 8.0 | 17.8 | **58.5** |
| Not resident | 0.9 | 0.3 | 0.8 | 0.8 | 0.2 | 0.1 | **3.1** |
| Tourist | 0.6 | 0.6 | 1.2 | 1.2 | 0.6 | 1.2 | **5.4** |
| Residency unknown | 3.2 | 1.7 | 5.1 | 4.2 | 2.3 | 16.4 | **32.9** |
| **Total** | **9.5** | **12.3** | **15.0** | **16.6** | **11.1** | **35.5** | **100** |
| AGE and VICTIM CONDITIONS | | | | | | | |
| | Child | Boy/girl | Young adult | Adult | Elderly | Age unknown | Total |
| Car | 1.2 | 2.7 | 4.7 | 4.0 | 1.0 | 2.3 | **15.9** |
| Other vehicle | 0.2 | 0.4 | 0.7 | 0.6 | 0.2 | 1.9 | **4.1** |
| Standing | 0.7 | 1.3 | 2.8 | 2.2 | 1.4 | 0.3 | **8.7** |
| Laying | - | - | 0.2 | 0.6 | 1.3 | - | **2.2** |
| Boat | - | 0.3 | 0.2 | 0.1 | - | 0.4 | **1.0** |
| Victim condition unknown | 10.1 | 4.8 | 6.3 | 9.2 | 7.2 | 30.6 | **68.1** |
| **Total** | **12.3** | **9.5** | **15.0** | **16.6** | **11.1** | **35.5** | **100** |
| AGE and VICTIM ACTIVITY | | | | | | | |
| | Child | Boy/girl | Young adult | Adult | Elderly | Age unknown | Total |
| Traveling | 1.4 | 3.1 | 4.7 | 4.0 | 1.4 | 3.6 | **18.2** |
| Working | 0.2 | 0.8 | 1.1 | 1.2 | 0.3 | 2.0 | **5.6** |
| Recreational activities | 0.5 | 0.8 | 1.4 | 0.8 | 0.1 | 0.8 | **4.4** |
| Sleeping | 0.2 | 0.1 | 0.2 | 0.6 | 1.2 | 0.2 | **2.5** |
| Rescuing someone | - | 0.4 | 0.7 | 0.4 | - | 0.1 | **1.7** |
| Victim activity unknown | 9.9 | 4.3 | 6.9 | 9.7 | 8.1 | 28.8 | **67.6** |
| **Total** | **12.3** | **9.5** | **15.0** | **16.6** | **11.1** | **35.5** | **100** |



385        **Table 11**. Age of flood fatalities crosschecked with accident place, accident dynamic, protective and hazardous behaviours.

| AGE and ACCIDENT PLACE | | | | | | | |
|---|---|---|---|---|---|---|---|
| | **Child** | **Boy/girl** | **Young adult** | **Adult** | **Elderly** | **Age unknown** | **Total** |
| A) Riverbed/riverside, Ford, Bridge | 1.9 | 2.3 | 3.6 | 3.1 | 1.0 | 2.5 | **14.4** |
| B) Road, Tunnel/underpass | 1.1 | 2.4 | 4.0 | 3.9 | 1.4 | 3.6 | **16.5** |
| C) Campsite/tent, Countryside, Bungalow, Recreation area | 0.9 | 0.6 | 0.5 | 0.8 | 0.6 | 0.4 | **3.7** |
| D) Public/private building | 2.7 | 1.1 | 1.9 | 2.6 | 3.9 | 5.5 | **17.7** |
| Place unknown | 5.7 | 3.1 | 5.0 | 6.2 | 4.1 | 23.6 | **47.6** |
| **Total** | **12.3** | **9.5** | **15.0** | **16.6** | **11.1** | **35.5** | **100** |
| AGE and ACCIDENT DYNAMIC | | | | | | | |
| | **Child** | **Boy/girl** | **Young adult** | **Adult** | **Elderly** | **Age unknown** | **Total** |
| A) Blocked in a flooded room | 1.0 | 0.4 | 0.9 | 1.3 | 2.6 | 1.1 | **7.4** |
| B) Caught in a bridge collapse, Caught in a road collapse, Fallen into the river | 1.0 | 1.2 | 1.8 | 1.8 | 0.8 | 1.0 | **7.6** |
| C) Dragged by water/mud | 7.8 | 5.4 | 8.0 | 7.7 | 3.3 | 16.9 | **49.1** |
| D) Caught in building collapse, Hit | 1.4 | 0.6 | 0.8 | 0.8 | 0.4 | 4.6 | **8.6** |
| E) Surrounded by water/mud | - | 0.2 | 0.5 | 0.3 | 0.2 | 0.1 | **1.4** |
| Dynamic unknown | 1.0 | 1.6 | 3.0 | 4.8 | 3.8 | 11.7 | **25.9** |
| **Total** | **12.3** | **9.5** | **15.0** | **16.6** | **11.1** | **35.5** | **100** |
| AGE and DEATH CAUSE | | | | | | | |
| | **Child** | **Boy/girl** | **Young adult** | **Adult** | **Elderly** | **Age unknown** | **Total** |
| A) Collapse/heart attack | 1.4 | 0.6 | 1.0 | 1.0 | 1.3 | 4.3 | **9.5** |
| B) Drowning | 9.8 | 8.0 | 12.3 | 13.3 | 8.5 | 16.7 | **68.5** |
| C) Hypothermia; Electrocution | - | - | - | - | 0.2 | - | **0.2** |
| D) Poly-trauma, suffocation, poly-trauma and suffocation | - | 0.2 | 0.7 | 0.4 | 0.6 | 0.4 | **2.3** |
| Death cause unknown | 1.1 | 0.7 | 1.0 | 2.0 | 0.6 | 14.1 | **19.4** |
| **Total** | **12.3** | **9.5** | **15.0** | **16.6** | **11.1** | **35.5** | **100** |
| AGE and PROTECTIVE BEHAVIOUR | | | | | | | |
| | **Child** | **Boy/girl** | **Young adult** | **Adult** | **Elderly** | **Age unknown** | **Total** |
| A) Climbing trees, grabbing on to someone/something | - | - | 0.2 | 0.2 | - | 0.1 | **0.4** |
| B) Driving to avoid danger, getting on the car roof, getting out of car | 0.3 | 0.4 | 0.8 | 0.7 | - | 0.3 | **2.6** |
| C) Getting out of buildings, moving to safer place, getting on roof/upper floor | 0.7 | 0.2 | 0.4 | 0.4 | 0.4 | 0.2 | **2.3** |
| Protective behaviour unknown | 11.3 | 8.8 | 13.6 | 15.4 | 10.7 | 34.9 | **94.7** |
| **Total** | **12.3** | **9.5** | **15.0** | **16.6** | **11.1** | **35.5** | **100** |
| AGE and HAZARDOUS BEHAVIOUR | | | | | | | |
| | **Child** | **Boy/girl** | **Young adult** | **Adult** | **Elderly** | **Age unknown** | **Total** |
| A) Check damage during flood | - | - | 0.2 | 0.2 | - | - | **0.5** |
| B) Driving on roads closed by police; Fording rivers | 0.6 | 0.6 | 1.6 | 1.0 | 0.3 | 0.1 | **4.2** |
| C) Refuse evacuation; Refuse warnings; | - | 0.3 | 0.2 | 0.4 | 0.1 | 0.4 | **1.5** |
| D) Staying on bridges; Staying on river banks | 0.4 | 0.5 | 0.6 | 0.6 | 0.2 | 0.3 | **2.7** |
| E) Trying to save vehicles; Trying to save belongings; Trying to rescue animals | 0.1 | - | 0.3 | 0.8 | 0.4 | 0.4 | **2.1** |
| Hazardous behaviour unknown | 11.2 | 8.1 | 12.0 | 13.7 | 10.0 | 34.2 | **89.1** |
| **Total** | **12.3** | **9.5** | **15.0** | **16.6** | **11.1** | **35.5** | **100** |



### 3.4    Trend in flood fatalities

In this section, the linear trend of some variables collected in EUFF 2020 for the 1980–2018 period, for either TOT-Area or individual areas, is analysed. According to the **number of FFs per FE**, the period studied has been divided in decades while FEs

has been divided in two groups: a) *low-severity FEs*, which caused the death of less than 10 people, and b) *high severity FEs*, which caused the death of more than 10 people (Table 12). The mean number of FFs per year shows the highest value between 1990 and 1999 (91), followed by the incomplete decade of 2010–2018 (61). In two remaining decades, these numbers were lower (48 in 1980–1989 and 54 in 2000–2009). In 14 out of 39 years, FFs were caused exclusively by low severity FEs, mainly in the 2000–2009 decade, during which on average 84% of FFs per year were caused by these events. In general, it seems that the total number

of FFs per decade is lower when the percentage of low severity events is higher (as in the 1980–1989 and 2000–2009 decades). Nevertheless, throughout the period studied, the proportions between low/high severity FEs did not show any evident trends.

Analysing the **trend of gender** of FFs at the scale of TOT-Area, the temporal trend of males FFs is stable, while the tendency of females FFs is clearly increasing, even though this varies among the studied areas (Figure 7). CAT and POR, according to the decreasing trend in FFs (Figure 4), show decreasing trends for both males and females FFs. The small increases in FFs trend in SFR

corresponds to increasing trend of females FFs, while the male trend seems to be stable. On the other hand, the increasing trends in GRE, CZE and ITA FFs is essentially due to an increase share of males FFs. In TUR, a slightly decreasing trend of both genders can be noted.

By the aforementioned data, the **trend of age** tends to move towards higher ages (Figure 8). FFs trend decreases for child and boy/girl, while it tends to increase in the other age classes. However, these tendencies must be analysed at the local scale, and

compared to the age of local populations that differs in structure (

Figure 5). For example, GRE and TUR show opposite age structure of population: the amount of elderly people is very large in GRE and very small in TUR. Nevertheless, a local analysis is not presented due to the scarcity of data in some of the areas studied.

**Table 12.** Proportions of flood events causing less than 10 flood fatalities (FEs with <10 FFs) and more than 10 FFs (FEs with >10 FFs) with respect to the total number of FFs per each year (Tot FFs/year) during the 1980–2018 period.


| Year | Tot FFs/year | FEs with <10 FFs | FEs with >10 FFs | Year | Tot FFs/year | FEs with <10 FFs | FE with >10 FFs |
|---|---|---|---|---|---|---|---|
| 1980 | 78 | 26.9 | 73.1 | 1990 | 97 | 41.2 | 58.8 |
| 1981 | 67 | 100 | - | 1991 | 91 | 47.3 | 52.7 |
| 1982 | 40 | 70.0 | 30.0 | 1992 | 84 | 50.0 | 50.0 |
| 1983 | 70 | 38.6 | 61.4 | 1993 | 36 | 100 | - |
| 1984 | 9 | 100 | - | 1994 | 97 | 53.6 | 46.4 |
| 1985 | 5 | 100 | - | 1995 | 174 | 22.4 | 77.6 |
| 1986 | 16 | 100 | - | 1996 | 70 | 100 | - |
| 1987 | 70 | 82.9 | 17.1 | 1997 | 112 | 58.9 | 41.1 |
| 1988 | 106 | 89.6 | 10.4 | 1998 | 95 | 38.9 | 61.1 |
| 1989 | 20 | 100 | - | 1999 | 49 | 77.6 | 22.4 |
| Tot. FFs | 481 | | | Tot. FFs | 905 | | |
| Mean | 48 | 80.8 | 38.4 | | 91 | 59.0 | 51.3 |

| Year | Tot | <10 | >10 | Year | Tot | <10 | >10 |
|---|---|---|---|---|---|---|---|
| 2000 | 46 | 71.7 | 28.3 | 2010 | 86 | 58.1 | 41.9 |
| 2001 | 47 | 100 | - | 2011 | 61 | 100 | - |
| 2002 | 100 | 54.0 | 46.0 | 2012 | 47 | 72.3 | 27.7 |
| 2003 | 37 | 100 | - | 2013 | 62 | 71.0 | 29.0 |
| 2004 | 29 | 100 | - | 2014 | 66 | 100 | - |
| 2005 | 52 | 100 | - | 2015 | 59 | 66.1 | 33.9 |
| 2006 | 79 | 55.7 | 44.3 | 2016 | 39 | 100 | - |
| 2007 | 32 | 100 | - | 2017 | 44 | 45.5 | 54.5 |
| 2008 | 22 | 100 | - | 2018 | 95 | 46.3 | 53.7 |
| 2009 | 93 | 54.8 | 45.2 | - | - | - | - |
| Tot. FF | 537 | | | Tot. FF | 559 | | |
| Mean???? | 54 | 83.6 | 40.9 | | 62 | 73.3 | 40.1 |

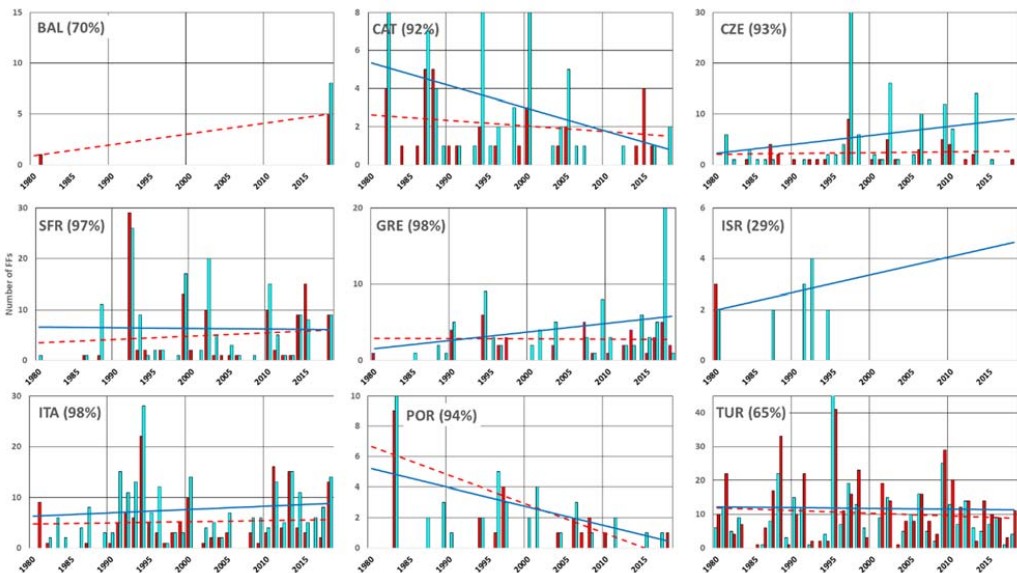

**Figure 7.** Linear trends in male and female FFs (flood fatalities) in each study area. Blue lines: linear trends in male FFs; red dashed lines: linear trends in female FFs.





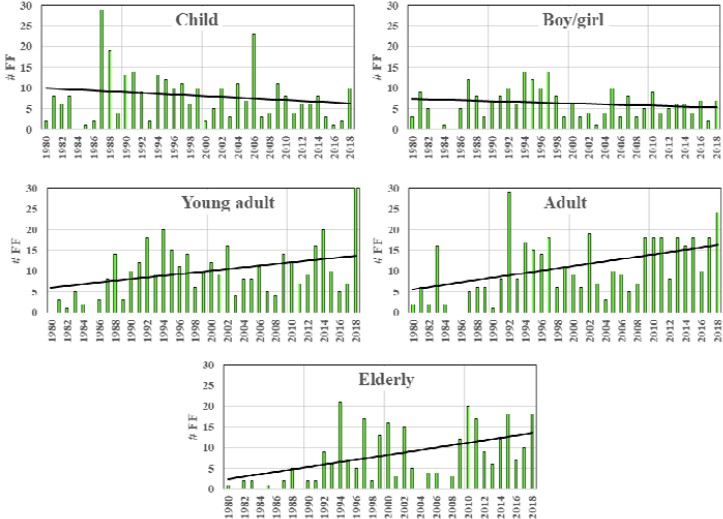

**Figure 8.** Linear trend in FFs (flood fatalities) for individual age classes in TOT-Area in the 1980–2018 period.

## 4  Discussion of the EUFF 2020 database potential

In order to enrich the EUFF 2020 database, data has been systematically collected from various documentary sources, widely used to record damages and casualties caused by natural hazards. Characteristics and limitations of the documentary sources have been extensively mentioned in the related literature (Brázdil et al., 2005; (Brázdil et al., 2006); Caloiero et al., 2014; Petrucci et al., 2019). Data completeness highly depends on the year of occurrence of FEs: i.e. less recent FEs are less documented due to the scarcity of information sources in the past years. On the other hand, FEs occurred more recently (the last two decades) are well documented due to a plethora of digital data sources, (e.g. news websites). Besides the aforementioned differences in data availability and sources, the inhomogeneity among the study areas must be also taken into consideration. Due to national privacy laws, the information collected for each FF may have a different level of detail depending on the study area and country.

The completeness of the database also depends on events' severity: FEs resulted in the death of one or a few persons could remain unnoticed or poorly documented in documentary sources, while FEs causing several people's death are usually much better documented and more in detail covered. This issue is more intense when data collection is performed at a global scale. For example, global disaster loss databases, as NATHAN (Natural Hazards Assessment Network) of the reinsurance company Munich Re, or the EM-DAT (Emergency Events Database) from the Centre for Research on the Epidemiology of Disasters of the Université Catholique de Louvain, have been proved useful for the identification of major events. However, these databases exclude FEs that caused the loss of a relatively low number of people: for example, in EM-DAT this number is 10. These thresholds are generally established based on severe disasters that happened in undeveloped countries, and thus would exclude the numerous fatalities due to FEs causing a few FFs, as it nowadays is frequently happening mainly in the Mediterranean countries (Llasat et al., 2013). By applying this criterion on EUFF 2020 database, 64.8% of FFs that occurred in the study areas during 1980–2018 (see Table 7) would have been missed.

Documentary sources are non-technical data sources: data obtained from documentary sources show unconventional format and are not measurable with standard criteria. However, this is the only kind of data useful to examine the impact of natural hazards such as floods (Barriendos et al., 2019) and other types of natural hazards such as earthquakes, tsunami (Alam, 2019), drought, heat waves (Camenisch et al., 2020), landslides (García-Garrido et al., 2020) or even to investigate past rainfall variability (Nash et al., 2016).



In order to avoid potential overlap (double counting of FFs), a thorough analysis of documentary sources has to be performed. Daily editions of newspapers published the following days of a FE often provide changing figures until the rescue operations have been finalized. Thus, just analyzing all the daily editions for days or weeks after a FE may allow the identification of the final numbers

of fatalities together with their accompanied profiles.

Despite all the potential sources of bias, in many countries, sources of documentation are still the only way to overcome the absence of systematic data collections as it in principle should be by local or national agencies. At European level, there are no "official" databases collecting FFs data for Member States. A first experiment, carried out in the Mediterranean environment by a multinational research group, was MEFF (MEditerranean Flood Fatalities) database, which included FFs occurred upon a 36-year period (1980–

2015) in five Mediterranean study areas (Aceto et al., 2017; Petrucci et al., 2019b; Vinet et al., 2019). Having MEFF as a starting point, we subsequently created EUFF 2019 by extending both the study area and the period, which was an added-value element in both spatial and temporal scales. The updated EUFF 2020 database improved data completeness of the previous version, as quoted in section 2.4., according to the main characteristics of documentary sources, which is their virtual "incompleteness". As it has been extensively quoted in the literature, research in documentary sources can never be considered as fully complete, since new sources

may become available or may be discovered with time. The availability of new data sources concerning old events can supply either unnoticed FEs or provide further details on FFs already existed. This is why the database, by using recently discovered data, has been significantly updated by e.g. increasing the number of FFs for CZE, including more attributes to FFs for SFR, or updating the list of FFs for GRE. Even if differences between absolute numbers of FFs is relatively small, it is important to incorporate these data on currently available variables on FFs. It must be also taken into account the importance of these variables in the in depth

understanding of what should be changed in order to substantially decrease flood risk for people. In fact, although for the severest FEs the number of FFs becomes known from newspapers or scientific and technical reports, additional useful information such as characteristics of floods' victims or details on the conditions under which the fatal accidents took place are not collected in any other database at the scale of the study areas. In general, details about the conditions and the characteristics of FFs are ephemeral: their knowledge mainly depends on the presence of potential witnesses or survivors and often is "buried-lost" due to privacy reasons.

Hence, each additional detail that may become available must be carefully collected and used to enrich further the data which can be the basis for further elaboration on the topic.

The novel element of this work, compared with the previous one (Petrucci et al., 2019), is that this work is focusing on trend analysis of data. In the frame of this publication we present for first time an analysis of number of casualties per flood events, which is a proxy of FE severity that can be compared to climatic trend. Besides, we also presented the temporal trend of gender (at local scale)

and age (at TOT-Area scale) of FFs. We present data in a disaggregated format, to allow other researchers easily detect specific characteristics of flood fatalities (i.e. age or gender) crosschecked with all the other variables that can be used in other kind of research. The simple data having been crosschecked by gender and age, allows easy comparisons with similar results already available in literature in different geographical frameworks (as i.e. in Australia and USA). Finally, the possibility to extract from the database data concerning a specific study area, ensures the possibility to go more in-depth from different points of view. In

particular, from sociological point of view, i.e., the examination of the relationships between FFs with demographic data and national development indices can be performed. From a hydrological-climatological-geomorphological point of view, the availability of year, month and day for all the 2483 FFs is a very consistent basis to set up a "threshold" analysis of rain that caused flood fatalities in the specific geomorphological framework of the study area. From a geographical point of view, due to the availability of coordinates of FFs, data can be used to draw the spatial distribution of flood mortality through a geographical information system,

as performed i.e. in Vinet et al., (2019) Actually, these hints seem very promising to improve knowledge on FFs in European countries. In our future plans it is the inclusion of more study areas, something which undoubtedly will enlarge and differentiate the geomorphological, hydrological, climatic and demographic framework in which relationships between local features and FFs could be carried out.



The first outcome of the current research concerns the gender of FFs. Even if, females are commonly considered as more vulnerable, in EUFF 2020 database female victims are less in absolute numbers than males, except for the elderly ones. The general concept of female's weakness during floods' events is affected by higher female's vulnerability in underprivileged frameworks (Zoleta-nantes, 2000). In South Asia, for example, the over representation of female victims depends on five features. 1) women are more likely to stay at home rather than evacuate to a shelter; 2) their dress restrict their movements; 3) cultural shame deters women from escaping to public areas if their clothing is ripped; 4) their inability to swim, which is also a consequence of cultural norms; and (5) being less well nourished (Yeo and Blong, 2010). Thus, gender alone is not a *de facto* driver of social vulnerability. However, it may rather become one, once it is correlated with age, occupation, access to health care (Alderman et al., 2012), and income. Take notice that low-income women experience the worst effects of flooding (Ajibade et al., 2013). Moreover, low-income population, regardless the gender, may suffer disproportionately human death and injury, as highlighted i.e. for color community in Texas from 1997 to 2001 (Zahran et al., 2008). The greater male vulnerability detected in EUFF 2020 can be related to a stronger exposure of males to floods, and to the higher proportion of males who drive vehicles (Jonkman and Kelman, 2005), due to either their wider mobility or outdoors working activities. Particularly, males are still more than females in outdoor works, and until recently, they outnumbered also in rescue services (e.g., fire fighters, police, and defense forces) (Salvati et al., 2012). Men are also more prone to exceed the standard safety rules, take more risks and to put themselves in danger e.g. to rescue people, belongings or pets.

In EUFF 2020, the majority of FFs concentrated on people of age between 30 and 64 years, thus in their most productive working years. It also explains the places and conditions of several FFs, who have been exposed to floods outdoor, while heading from home to work premises or vice-versa. On the contrary, elderly (retired) people have been more frequently affected indoor while being at home (Diakakis et al., 2020). Thus, elderly people are more frequently trapped by flood in their home, while adults and children are dragged outdoors (Haynes et al., 2015). In contrast to related studies (Zoleta-Nantes, 2000), our work did not detect any particular vulnerability features among children. In our study, it seems that the trend of young victims throughout the study period is decreasing.

*Car* or *other vehicles* are found to be the most frequent condition of victims in each study area, for both males and females, as extensively stated in literature, even though our data did not allow to test the interesting suggestion according which outdoor incidents are more abundant in non-urban environments (Diakakis et al., 2020).

Regarding protective and hazardous behaviors, our data are scarce, but they do confirm opinions accepted in literature according which males are more prone to risk by taking unnecessary actions (e.g., Salvati et al., 2018), even if we did not detected actions influenced by drugs or alcohol, as some authors highlighted (Franklin et al., 2017). Compared to males, females behaved in a less hazardous way.

For 28.1% of FFs, even if we know the exact time of the accident, we did not examine the potential effect of this variable. In principle, based on the hour of the accident, we could identify light conditions at the moment of the accident and try to potentially correlate it with the way it affected the accident's development, as is also investigated in the related literature (e.g. Špitalar et al., 2014). Because of the location of the study areas in different latitudes, obtaining light conditions by hours it is needed to know the time of sunset and sunrise according to corresponding local time, a study element which was not on the frame of the current work. Finally, our data series did not confirm the decreasing trend in FEs with multiple fatalities and the increasing trend in FEs with only a few fatalities (e.g., Diakakis, 2016; Pereira et al., 2017). These two different severity levels in FEs continue to both occur during the entire studied period without any particular trend, even if TUR seems to be the most frequently affected by FEs resulted in the death of more than 10 people.

Finally, we would like to underline that the term "European" is ambitious given that EUFF 2020 deals with only eight countries. Adding further study areas-countries is not a trivial task and an easy-to-do survey, especially nowadays that this kind of research has no dedicated funding. If we wish to add a new study area from scratch, a systematic survey of local data sources concerning 39 years has to be performed. This work must be necessarily performed by native researchers, since they can rapidly, easily and effectively analyse the huge amount of sources in which the requested information on FFs can be extracted from. Hence, researchers



who already use documentary sources to study natural hazards may perform this work easier. Researchers that already collected documentary sources depicting the historical series of floods in their country could straightforward re-analyse original sources to extract data on FFs, and eventually look for lacking details in coeval data sources. For all the aforementioned reasons, new study areas can not be selected based, for example, on geographical criteria (i.e. select areas at a certain latitude or longitude). Our aim is to take into account all opportunities and willingness to find and collaborate with even more future partners towards common goals.

## 5     Data availability

*EUropean Flood Fatalities (EUFF) database 1980-2018 (updated)* is available in the 4TU Centre for Research Data (Petrucci et al., 2020, https://doi.org/10.4121/uuid:489d8a13-1075-4d2f-accb-db7790e4542f, 2020). The EUFF 2020 database collects 2483 FFs that occurred during a 39-year period in 9 study areas. It includes three files: a) a *comma-separated values* (csv) file, which contains the data; b) the *keyhole mark-up language* (kml) file, which provides the location of fatalities on the Google Earth; and c) the *readme* (txt) file, containing the description of database structure.

## 6     Conclusions

In the current work, the *EUFF 2020* database has been introduced recording data on flood fatalities for 9 areas of 8 European countries for the 1980–2018 period. The potential of the database to analyse various aspects of the structure of the victims due to flood events is also shown. Highlighted conclusions can be shortly summarised as follows:

(i)     The studied European countries follow various practices for the data collection of flood fatalities on national or regional levels. Similarly, various sources of documentary evidence they do exist. For example, Media information (newspapers, TV, internet reports) have been among the main sources of such records in the past decades.

(ii)     Despite the limited documentary sources, mainly concerning underreporting issues, bias towards the most severe events and inhomogeneity from one country to another, *EUFF 2020* shed light on the fatal flood events occurred in the eight studied countries of the Euro-Mediterranean region.

(iii)     The *EUFF 2020* database provides both regional and super-regional analyses of floods fatalities concerning their gender, age, conditions, activity of fatalities and dynamics of the accidents, thus contributing to a better understanding of the human exposure to floods associated with the most serious potential consequence, which is its own death.

(iv)     The EUFF 2020 database, with its great potential to be extended spatially and temporally, represents a unique European database of high scientific and practical potential. One aspect of the database is its vitality, as we continuously strive to improve it by extending the study period and enlarging the domain. The EUFF 2020 database and its high potentials will hopefully motivate and encourage more researchers to enrich it with data on FF available in their own countries.

Further EUFF 2020 spatial and temporal extension may allow the comparison of different local frameworks in a broader European scale and the identification of general and local features useful in risk management and educational campaigns. This will subsequently allow the assessment of climate change effects, differences in demographics, economical and the technological developments on these fatalities and their temporal/spatial variability. We believe that the followed pan-European approach, not only frames the anticipation of flood fatality risk into a broader context but also promises benefits for diverse scientific disciplines.

Floods and its fatal consequences can be mitigated more efficiently by developing a holistic view, by building safety measures with sharing useful experiences, best practises and lessons learned among European countries. To this direction, EUFF 2020 may contribute to outline public policies and civil protection campaigns, reduce the impact of floods and hopefully minimize the number of future fatalities.





## 7    Acknowledgements

RB and JŘ were funded by the Ministry of Education, Youth, and Sports of the Czech Republic for the SustES – Adaptation strategies for sustainable ecosystem services and food security under adverse environmental conditions project, ref. CZ.02.1.01/0.0/0.0/16_019/0000797.

MCLL and MLL were funded by the Research State Agency of the Ministry of Science and Innovation and the Ministry of Universities under the National Project M-CostAdapt (CTM2017-83655-C2-2-R).


Thanks to Dr. Theodoros Baimpos for proofreading the article.

## 8    Author Contributions:

Conceptualization: O.P. Original manuscript written by O.P. Database management: L.A., M.M. Validation: R.B., S.P., K.P. Data collection: C.B., V.B., M.I., Ö.K., M.L., J.Ř, J.R.G., P.S., V.K., J.L.Z. Manuscript review: A.K., F.V., M.C.L.


Funding: This research received no external funding.



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
