# Peer review of "EUFF (EUropean Flood Fatalities): A European flood fatalities database since 1980"

_Earth System Science Data, 2020_

## Referee Comment (RC1) · Dominik Paprotny (Referee) · 17 Oct 2020

Dear authors,

Your paper "EUFF (EUropean Flood Fatalities): A European flood fatalities database since 1980" cover an important topic. Easily accessible and consistent databases of fatalities are rare and definitely needed to advance the understanding of the process. That being said, I see two major issues with your submission: the overlap with the two previous papers (Petrucci et al. 2019a and 2019b, op. cit.) and the overall structure of the paper. Both points are somewhat related. I think the authors themselves summarize the issue best: "[t]he novel element of this work, compared with the previous one (Petrucci et al., 2019), is that this work is focusing on trend analysis." An ESSD paper

should focus on the dataset in terms of methods and validation rather than results. In this context my main methodological concerns are:

1) Defining a flood fatality: given the centrality of the concept, it should be more strictly defined and discussed. For instance: do people missing and presumed dead count as well (as in the example, L125)? Do deaths from injuries suffered in the event count if they occurred at some point after the event? What about deaths from compound events comprising both floods and landslides? Are all types of floods included (coastal, fluvial, pluvial)? When is a fatality "indirect"? The database makes such no distinction, but I imagine that an "indirect" fatality would be e.g. result of evacuation of vulnerable people as in the Var river flood in 2010.

2) No discussion is made on the reliability of the sources involved. For instance, the example presented in section 2.1 highlights, news reports made while the event was still unfolding might not be reliable. This could be partly responsible for differences in fatalities reported in different databases. Do the authors consider the database being complete for individual flood events? Could they be used instead of EM-DAT, Munich Re or other resources in this matter? Is the dataset an improvement compared to those datasets (other than including smaller events, as noted in section 4)?

3) The authors write that "[t]he analysis of newspapers is a long process. It requires the selection of several articles...". How did the process look like? Did it start with a list of dates and locations of flood events from some external source, or every single issue of a given newspaper was scoured for any mention of floods?

4) No exhaustive list of sources is presented (despite many types of those mentioned) nor the individual records have an indication of the source (compare e.g. the detailed listing of every single piece of information in Paprotny et al. 2018a). What is the relationship between "Existing local database" in Table 1 and EUFF? Are they transferred 1-to-1 to EUFF or also the authors' additional research (as suggested in lines 154-155)? Also, what was the procedure of improving the completeness of the database

(section 2.4)? Were the original sources revisited to extract more data on variables not previously considered, or new sources were discovered?

5) The authors include variable "FLOOD EVENTS" in Table 4 and mention flood events multiple times, but there is no identifier or other link to flood events in the database. Where does the number of flood events come from? Are those simply unique dates and e.g. region combinations? I couldn't arrive to the number presented in Table 4 after trying various combinations of variables. Also, is there a reason why information about the events themselves is not recorded at all, e.g. the type of flood? This would have been very useful in analysis of the fatalities, for those interested in, say, coastal floods (Bouwer and Jonkman 2018).

6) The discussion and conclusions mention e.g. "recent FEs are less documented due to the scarcity of information sources in the past years "; "differences in data availability and sources, the inhomogeneity among the study areas must be also taken into consideration"; "limited documentary sources, mainly concerning underreporting issues", but none of those aspects is elaborated. It has, however, potentially very strong influence on any trend analysis of fatalities. Did the authors consider analyzing the completeness of the database in some way (see e.g. Paprotny et al. 2018b)?

7) Finally, regarding Table 2 and the dataset proper: it's a bit strange to record the lowest administrative unit as "prefecture" as it is a very particular type of unit existing only in some countries. The recording of different levels of regions, municipalities and "prefectures" doesn't appear very consistent. A table explaining which administrative units are recorded for each country in the study would be helpful. Also, I would recommend for future development of the database to add some kind of identifier connecting EUFF records with existing geocoding systems such as NUTS for regions or LAU for municipalities (where possible, but it definitely is at least the EU member countries).

The paper contains a lot of results, further repeated in the discussion with strong focus on analyzing trends and causes of deaths (about two-thirds of the paper). This is

in fact heavily repetitive from two preceding papers, even if organized in tables and figures somewhat differently (though Fig. 1 is taken, uncited, directly from Petrucci et al. 2019a). Decision on whether there is, in fact, too much overlap or too much focus on results belongs of course to the editor, but I would strongly recommend rebalancing the paper to provide sufficient methodological detail (as listed above) and cut down the results significantly, as they would be too detailed even for a "regular" research paper. I think the database will be a valuable resource and I am really impressed by the authors' effort in creating EUFF, but it deserves a better documentation than it currently has. I am confident that the authors' would be able to revise it accordingly, as almost all issues mentioned in this review pertains only to the composition of the paper.

Kind regards,

Dominik Paprotny

Some minor things noted down while reading the manuscript:

L29-30: duplicated text

L30, 33, 108: Petrucci et al. 2020 is not listed in the references. Is this the reference aligned with the link? Anyhow it should be removed from the abstract.

The abstract contains almost exclusively results, whereas the methodology and contents of the database should be the main focus.

L57: "the resulted increasing concentration..." – the sentence suggests (wrong grammar notwithstanding) that increase in exposure is a result of climate change, which is not true.

L101: the paragraph should start with something like "The paper is organized as follows:"

Table 1: I would suggest to some more reliable resource for demographic data here and further in the study than "www.Worldometers.info" or "http://www.populationpyramid.net/" (Fig. 5) e.g. Eurostat and other statistical agencies.

Table 1: The composition of "South France" should be explained in the text, not only as a table caption, which makes it easy to miss.

Fig. 1: the picture is exactly the same as in the other paper, not cited. Also, there is no reference to the source of the underlying satellite image.

Table 2: hour as time and string, and hour accuracy as text and integer? I think string and integer are switched? Also, if the table indicates conversion of time-of-day descriptions into hours, then where does e.g. 1530 hrs fit in that case?

Table 3: two columns named "description"

Table 7. "Number of flood fatalities (FFs) per flood event (FE) as percentage of total FFs." -> I think you mean FFs per severity class of FE. Similarly confusing in Figure 3.

L457-458: those methodological aspects are not mention in the methods section.

L420, 467: incorrect citation.

References

Paprotny D., Morales Nápoles O., Jonkman S.N. 2018a. HANZE: a pan-European database of exposure to natural hazards and damaging historical floods since 1870. Earth System Science Data 10:565–581, doi:10.5194/essd-10-565-2018.

Paprotny D., Sebastian A., Morales Nápoles O., Jonkman S.N. 2018b. Trends in flood losses in Europe over the past 150 years. Nature Communications 9:1985, doi:10.1038/s41467-018-04253-1.

Bouwer L. M. Jonkman S.N. 2018. Global mortality from storm surges is decreasing. Environmental Research Letters 13(1):014008, doi:10.1088/1748-9326/aa98a3

---

## Referee Comment (RC2) · Jonathan Gourley (Referee) · 18 Nov 2020

Reviewer Summary:

This article describes a flood fatality database available for 8 European countries for a time period from 1980-2018. The database includes a number of variables that can be pieced together to help describe the specific circumstances surrounding each fatality event. Many of the details were extracted by newspaper reports and thus requires a great deal of manual work to ascertain the details. The database is thus quite useful for research purposes and practical applications. I was able to successfully download the database and could interpret the variables quite easily. I did note that the file was described as csv, or comma-delimited, but the fields were delimited by semi-colons.

[Figure]

Anyhow, not a big deal, but there are some major points that need to be addressed prior to publication. I highlight these below.

1. Novelty of the manuscript - The EUFF 2020 database is an update to the EUFF 2019 one that was described in Petrucci et al. (2019). EUFF 2020 provides more data from prior events and includes more geographic regions, totaling 17 new flood fatalities. I note that this is still far from including the total land area of Europe. Nevertheless, the information presented in the new article is very similar to that of Petrucci et al. (2019), the latter of which uses more figures than data in tables presented in the current article. I preferred the presentation of the data in the form of figures as shown in Petrucci et al. (2019). But, I see very similar data being used in both articles. I note that the authors address this potential shortcoming by stating that the new paper includes a trend analysis of the fatalities, broken down into age and gender categories. The issue I see with this is they are already working with a small sample size of data. Once it's further segregated by country, year, age, gender, etc., then it gets even smaller. I'm not comfortable with fitting lines to the small samples of data to ascertain if there is a significant trend. I don't have as much experience reviewing articles describing databases, but if this were a "normal" manuscript, then I would not consider this to be novel enough for a stand-alone publication. Nevertheless, this is more of a decision for the topical editor.

2. Presentation quality - I would strongly suggest that the authors improve the general readability of the manuscript. I began making notes about improper grammar beginning with the first sentence in the abstract. I quit making notes on the grammar because they were quite numerous. I also note that the presentation of the article text comes across more as an outline or draft rather than a manuscript in final form. There are lots of paragraphs containing a single sentence, improper indenting, and use of boldface in the text as one might do in a proposal, etc.. I would expect this for materials that are being presented orally, but not in a written, published format.